# Gating mechanisms during actin filament elongation by formins

**Fikret Aydin[1,2,3], Naomi Courtemanche[4,5], Thomas D Pollard[5,6], Gregory A Voth[1,2,3]***

[1]Department of Chemistry, The University of Chicago, Chicago, United States; [2]Institute for Biophysical Dynamics, The University of Chicago, Chicago, United States; [3]James Franck Institute, The University of Chicago, Chicago, United States; [4]Department of Genetics, Cell Biology and Development, University of Minnesota, Minneapolis, United States; [5]Department of Molecular, Cellular and Developmental Biology, Yale University, New Haven, United States; [6]Department of Molecular Biophysics and Biochemistry, Yale University, New Haven, United States

**Abstract** Formins play an important role in the polymerization of unbranched actin filaments, and particular formins slow elongation by 5–95%. We studied the interactions between actin and the FH2 domains of formins Cdc12, Bni1 and mDia1 to understand the factors underlying their different rates of polymerization. All-atom molecular dynamics simulations revealed two factors that influence actin filament elongation and correlate with the rates of elongation. First, FH2 domains can sterically block the addition of new actin subunits. Second, FH2 domains flatten the helical twist of the terminal actin subunits, making the end less favorable for subunit addition. Coarse-grained simulations over longer time scales support these conclusions. The simulations show that filaments spend time in states that either allow or block elongation. The rate of elongation is a time-average of the degree to which the formin compromises subunit addition rather than the formin-actin complex literally being in 'open' or 'closed' states.

DOI: https://doi.org/10.7554/eLife.37342.001

*For correspondence:
gavoth@uchicago.edu

**Competing interests:** The authors declare that no competing interests exist.

## Introduction

Actin is one of the most abundant proteins in eukaryotic cells and is important for numerous functions regulated by interactions with many other proteins (*Pollard and Cooper, 2009*; *Dominguez and Holmes, 2011*; *Blanchoin et al., 2014*). The dynamic actin network in the cell cortex enables cells to maintain specific shapes in addition to supporting the plasma membrane. Actin filaments in lamellipodia drive cellular movements, and contractile rings of actin filaments are responsible for cytokinesis (*Chesarone et al., 2010*). Transitions of actin between monomeric and filamentous states is a key feature of these dynamic systems (*Pollard and Cooper, 2009*).

Formins regulate actin assembly by nucleating and directing the elongation of unbranched actin filaments (*Grikscheit and Grosse, 2016*). Formin malfunctions are associated with cancer (*Favaro et al., 2006*; *Favaro et al., 2003*; *Lizárraga et al., 2009*; *Sahai and Marshall, 2002*) and immune disorders (*Colucci-Guyon et al., 2005*), so characterizing formins is crucial for understanding these important diseases.

Formins consist of multiple domains, including a conserved formin homology 2 (FH2) domain that self-associates in a head-to-tail manner to form a homodimer that stabilizes actin filament nuclei (*Figure 1*). FH2 dimers stay processively associated with the filament's barbed end by 'stepping' onto actin subunits as they incorporate into the growing filament. FH1 domains are located directly N-terminal to the FH2 domain and consist of flexible regions connecting multiple polyproline tracks that

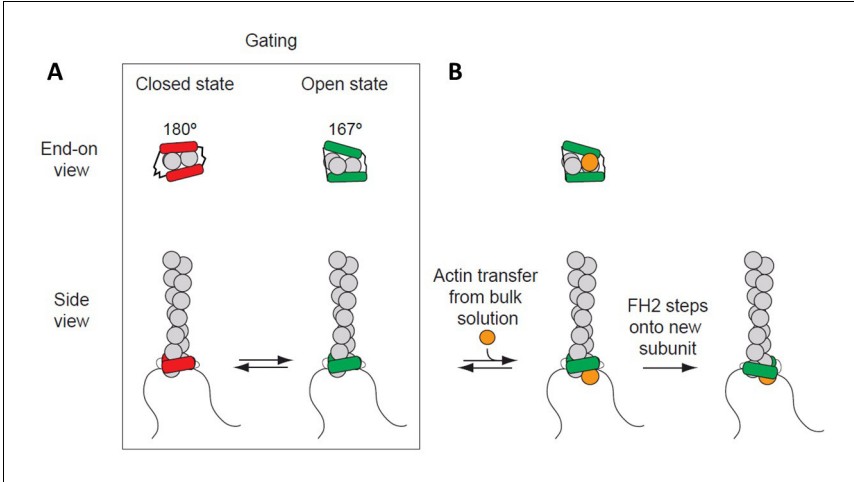

**Figure 1.** Processive association of formin with the barbed end of a growing filament. (**A**) End-on and side views of formin FH1FH2 domains interacting with the barbed end of an actin filament (gray) in closed (FH2 domain in red color) and open (FH2 domain in green color) states. The black curved lines represent FH1 domains. (**B**) The addition of an actin subunit (orange) from the bulk solution to the filament end, and stepping of an FH2 domain (green) onto the newly added actin subunit.

DOI: https://doi.org/10.7554/eLife.37342.002

each bind a profilin-actin complex. Diffusive motions of the FH1 domains allow actin to transfer rapidly onto the barbed end.

Actin filament barbed ends associated with a formin FH2 domain elongate slower than free barbed ends (*Paul and Pollard, 2009a*; *Kovar et al., 2006*). This effect is considered to arise from 'gating,' a rapid equilibrium between an 'open state' (when an actin subunit can bind the barbed end) and a 'closed state' (when the barbed end is blocked) (*Paul and Pollard, 2009a*; *Kovar et al., 2006*; *Vavylonis et al., 2006*). The fraction of time that a barbed end is in an 'open' state is defined as the gating factor, which ranges from ~0.95 for mammalian formin mDia1 to ~0.5–0.7 for budding yeast formin Bni1 and to ~0.05 for fission yeast formin Cdc12 (*Kovar et al., 2006*; *Harris et al., 2006*; *Moseley and Goode, 2005*). Thus, barbed ends with mDia1 are mostly in an 'open' state that does not slow elongation, while barbed ends with Cdc12 are mostly closed, strongly inhibiting elongation. Rapid transfer of actin from FH1 domains onto open barbed ends allows filaments to elongate rapidly, in spite of gating (*Chang et al., 1997*; *Courtemanche and Pollard, 2012*; *Paul et al., 2008*; *Watanabe et al., 1997*).

Two mechanisms are proposed for formins to gate the barbed end (*Paul and Pollard, 2009b*; *Otomo et al., 2005*; *Pring et al., 2003*; *Xu et al., 2004*; *Goode and Eck, 2007*). In a closed state, the formin might sterically interfere with the incoming actin subunit and/or distort the barbed end of the filament to compromise binding. An open state is assumed to have no steric interference from the FH2 domain to an incoming monomer binding between the terminal and penultimate subunits (A2 and A3 in the nomenclature in *Figure 2B*). Some (*Paul et al., 2008*; *Paul and Pollard, 2009b*) but not all (*Otomo et al., 2005*; *Goode and Eck, 2007*) models assume that an open end of the filament has the natural 167° helical twist that is favorable for adding a subunit. Most models assume a rapid equilibrium between open and closed states but differ in detail. The original crystal structure of the Bni1 FH2 dimer (*Xu et al., 2004*) suggested that the trailing FH2 domain sterically blocks subunit addition. The structure of Bni1 FH2 domains bound to actin with a 180° twist (*Otomo et al., 2005*) revealed that FH2 domains can not only interfere sterically with elongation but also distort the filament into flattened helical twists > 167° that are unfavorable for subunit addition. Our study addresses the contributions of steric blocking and helix distortion to gating.

A separate issue closely related to gating is how FH2 domains move on the barbed end as the filament elongates. In 'stair-stepping' models (*Otomo et al., 2005*; *Xu et al., 2004*; *Goode and Eck, 2007*) one FH2 domain remains associated with the terminal and penultimate subunits (A2 and A3 in in *Figure 2B*), while the other FH2 domain alternates between two positions. When bound to the A3

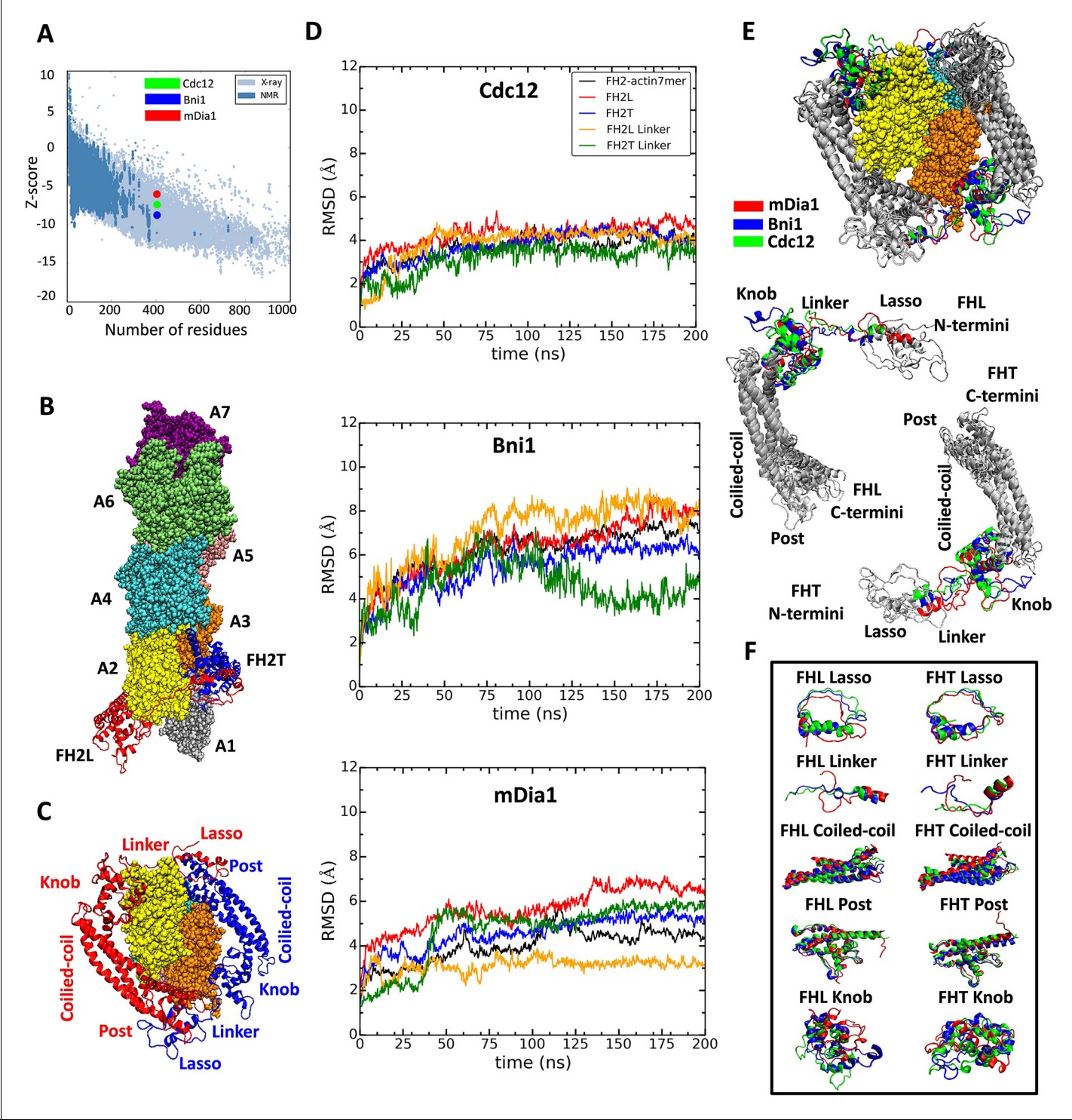

**Figure 2.** Homology models of FH2 domains on the barbed end of actin filaments. (**A**) Overall quality of the homology models of Cdc12, Bni1 and mDia1 FH2 domains after 200 ns of MD simulations. The graph compares the z-scores of the three homology models with the z-scores of all experimentally determined native proteins from Protein Data Bank. (**B, C**) Ribbon diagrams of a dimer of Cdc12 FH2 domains interacting with the barbed end of a space-filling model of an actin filament seven-mer in the state before the formin steps onto the newly added actin subunit A1 on the barbed end. The actin subunits are numbered from A1 to A7, starting from the barbed end. (**B**) Side view. (**C**) View from the barbed end without subunit A1 and with labels on the regions of FH2 domains. (**D**) The root mean square deviations (RMSD) of C-alpha atoms over time during the all-atom MD simulations of dimers of FH2 domains of Cdc12, Bni1 and mDia1 on an actin seven-mer. The first 160 ns of the trajectory for Bni1 is from *Baker et al. (2015)*. The trajectories of the whole complex, FH2 domains and linkers are displayed in different colors. (**E, F**) Structural alignment of
*Figure 2 continued on next page*

*Figure 2 continued*

ribbon diagrams taken at the end of 200 ns all-atom simulations of the FH2 domains of Bni1, Cdc12 and mDia1 associated with the actin filament seven-mer. Structural features are labeled. (E) Views from the barbed end with actin subunits A2 and A3 in the upper panel. The FHL and FHT domains are shown separately by aligning the coiled-coil regions in the lower panel. Superimposed features are shown in gray and features that differ between the three formins are color-coded. The actin filaments are not aligned and only shown to guide the location of formins. (F) The superimposed ribbon diagrams of the separate parts of the FH2 domains of the three formins.

DOI: https://doi.org/10.7554/eLife.37342.003

The following source data is available for figure 2:

**Source data 1.** Cdc12 FH2 interacting with a seven-mer filament.
DOI: https://doi.org/10.7554/eLife.37342.004
**Source data 2.** Bni1 FH2 interacting with a seven-mer filament.
DOI: https://doi.org/10.7554/eLife.37342.005
**Source data 3.** mDia1 FH2 interacting with a seven-mer filament.
DOI: https://doi.org/10.7554/eLife.37342.006
**Source data 4.** Cdc12 FH2 interacting with a five-mer filament.
DOI: https://doi.org/10.7554/eLife.37342.007
**Source data 5.** Bni1 FH2 interacting with a five-mer filament.
DOI: https://doi.org/10.7554/eLife.37342.008
**Source data 6.** mDia1 FH2 interacting with a five-mer filament.
DOI: https://doi.org/10.7554/eLife.37342.009

and A4 subunits, this trailing FH2 domain distorts the helix and interferes with subunit addition. When the trailing FH2 steps forward, its knob binds subunit A2 and its post dissociates from the barbed end to open the end for subunit addition. Stair-stepping models assume that the dissociated FH2 does not interfere sterically and that the incoming subunit can bind the flattened end. The 'stepping second' model (*Paul et al., 2008*; *Paul and Pollard, 2009b*) proposes that both FH2 domains remain bound to subunits A2, A3 and A4 during a rapid equilibrium between open states with a helical twist close to 167° and little or no steric blocking and closed states with steric blocking and/or unfavorable helical twists. When an end is open, a new subunit can bind, followed by rapid, reliable stepping of the trailing FH2 domain onto the new subunit. An interesting variation of stepping second assumes that the formin dimer transitions from being bound to the three terminal subunits to binding only the two subunits closest to the barbed end, prior to subunit addition and final displacement of the trailing FH2 onto the new subunit (*Thompson et al., 2013*). A fourth model assumes that the pair of FH2 domains moves in a screw-like fashion along the short-pitch helix of the elongating actin filament (*Shemesh et al., 2005*). The relationship of this mechanism to gating is unclear.

Understanding the mechanism of gating is important, because gating influences subunit addition from both solution and from FH1 domains (*Kovar et al., 2006*; *Vavylonis et al., 2006*; *Paul et al., 2008*; *Neidt et al., 2008*; *Courtemanche et al., 2013*; *Gurel et al., 2015*). Given the large body of experimental work without mechanistic tests, we sought to identify the gating mechanism of formins by examining three formins with different gating factors, mDia1, Bni1 and Cdc12. We used both all-atom (AA) and coarse-grained (CG) molecular dynamics (MD) as well as enhanced free-energy sampling simulation techniques to examine the interactions between the FH2 domains of these formins with an actin filament. We found that both steric blocking and distortion of the barbed end can contribute to gating with implications for both the 'stair-stepping' and 'stepping second' models of processive elongation.

## Results

### Homology models of FH2 domains associated with actin filament barbed ends

The first step in our comparison of the three formins was to build homology models of Cdc12 and mDia1 FH2 domains bound to actin, for which no structures are yet available. We based the new models on the only detailed model of FH2 domains bound to an actin filament barbed end

(*Baker et al., 2015*). This model of the Bni1 FH2 domain at the barbed end of a filament consisting of seven actin subunits was based on a co-crystal structure of Bni1 FH2 and actin with a twist angle of 180° (*Otomo et al., 2005*) and an X-ray fiber diffraction model of the actin filament with a twist angle of 167° (*Oda et al., 2009*). *Baker et al. (2015)* used the intermolecular contacts in the crystal structure to place the FH2 domains on the 167° barbed end and refined the model with 160 ns of AA MD simulations, during which conformational changes closed the contacts between the FH2 domains and actin, and flattened the helical twist at the barbed end of the filament. A 3.4 Å crystal structure of a dimer of FMNL3 FH2 domains bound to actin has contacts between the FMNL3 FH2 domain and actin similar to those of the Bni1-FH2 domains (*Thompson et al., 2013*). However, this structure has a two-fold axis of symmetry between physically separated actin subunits, so it is less informative regarding the structure of FH2 domains bound to a helical actin filament than the Bni1-FH2-actin structure, upon which *Baker et al. (2015)* and we based our models for MD simulations.

We created homology models of Cdc12 and mDia1 FH2 domains based on the crystal structure of the Bni1 FH2 domain and used Baker's model of the Bni1 FH2-actin filament (*Baker et al., 2015*) to align these FH2 models on the end of a filament consisting of seven subunits (see the Materials and methods section for details). The FH2-actin complexes were solvated, ionized and then refined by extensive AA MD simulations (*Baker et al., 2015*). These models with an FH2 dimer on subunits A2, A3 and A4 of a seven-subunit filament correspond to the conformation immediately after the addition of a new actin subunit A1. We used 200 ns AA simulations to assess the quality and stability of the homology models, to document the contacts between FH2 domains and actin, and to determine differences between the three FH2 domains interacting with actin. After 200 ns of simulation, we created structures corresponding to the step before the addition of subunit A1 by removing subunit A1 from the barbed ends of the models of the three seven-mer filaments. For computational efficiency, we also removed actin subunit A7 at the pointed end, leaving five actin subunits in the filament. The FH2 domains are associated with the same subunits in the five-mer model and the Otomo 'blocked, n state' (*Otomo et al., 2005*). This state is assumed to be blocked in the crystal structure of Otomo et al., so the trailing FH2 domain must step off the end of the filament to the 'accessible n state' before a new subunit can bind and return the end to the 'blocked, n + 1 state.' We did not simulate the 'accessible n state.' We extended the AA simulations of seven-mer and five-mer filaments to assess steric interference between the FH2 dimers and actin subunits binding to the barbed end of the filament and to determine how the FH2 domains influence the geometry of the terminal actin subunits. Coarse-grained (CG) simulations of the FH2-actin models allowed us to study deviations from their initial configurations at time scales beyond the current range of the AA simulations.

## Evaluation of the models of FH2 dimers on actin filament seven-mers after all-atom MD simulations

At the end of the 200 ns simulations, the total energies of the three models of formin dimers interacting with actin filaments (calculated by ProSA [*Wiederstein and Sippl, 2007*]) were within the range of protein structures in the PDB determined by X-ray crystallography and NMR (*Figure 2A*). The z-scores indicate that the Bni1 model (z-score = −8.9) is the most accurate (as it is based on a crystal structure), and Cdc12 (z-score = −7.69) is more accurate than mDia1 (z-score = −6.45), because the sequence of the Bni1 template model is more similar to the Cdc12 model than the mDia1 model. The z-score is calculated by taking both structural features and the sequences of FH2 domains into account. Forcing the FH2 primary sequence to fold into a structure other than an FH2 domain will give a z-score outside the range of high-quality structures.

The models of the three formin/actin systems obtained from the AA simulations are very similar (*Figure 2E*). All three models have the FHL (leading FH2 domain) and FHT (trailing FH2 domain) positioned head-to-tail to form a donut-like shape around the barbed end of an actin filament with seven subunits (*Otomo et al., 2005*). *Figure 2B* shows the FH2 domain of Cdc12 interacting with the barbed end of an actin filament with the FHL domain engaged with actin A2 and the FHT domain engaged with actin A3, prior to FHT stepping onto the newly added actin subunit A1 at the barbed-end. Each FH2 subunit consists of helical knob and post regions connected by a three-helix bundle (coiled-coil). The N-terminal lasso region of each subunit encircles the post of the other subunit and is connected to the knob of its own subunit by a flexible linker (*Figure 2C*). The coiled-coil and post regions of the FH2 domains of the three formins align quite well (*Figure 2E*), while other

regions differ locally (*Figure 2F*). The lasso regions of the FH2 domains have similar circular conformations, but the mDia1 FH2 lasso lacks a helix found in Cdc12 and Bni1. The helical parts of the linker regions align quite well with each other, but the unstructured parts differ in length and conformation. The knob regions align the least well. The mDia1 knob has fewer helices than the Cdc12 and Bni1 knobs. Also, the locations of the helices and loops in the mDia1 knobs differ much more than those in the knobs of the other two formins.

## Do FH2 domains interfere sterically with the addition of an actin subunit at the barbed end?

We used AA simulations to test the steric blocking hypothesis for gating. The Baker et al. simulations (*Baker et al., 2015*) of Bni1 brought the FHL and FHT domains into much closer contact with the actin subunits at the barbed end including small overlaps of the FHL domain with incoming actin subunit +A1 placed on the barbed end of the filament with the Oda filament (*Oda et al., 2009*) geometry. We measured the volume fraction of this overlap by determining which C-alpha atoms of FHL and actin subunit +A1 were separated by <4 Å (the diameter of a C-alpha atom). The models of the other two formins were based on the Bni1 model after 160 ns of simulation (*Baker et al., 2015*), so their FH2 domains overlapped the +A1 site to the same extent as Bni1 at the beginning of the simulations (*Figure 3A,B,C*). During the remainder of the 500 ns AA simulations, steric interference with subunit +A1 increased for Cdc12 and decreased for mDia1, with Bni1 in between (*Figure 3A*).

During two rounds of AA simulations of five-mer filaments lacking subunit A1, the steric interference of FHT with the site where subunit A1 binds increased to high levels for Cdc12 and fluctuated between none and low levels for Bni1 and mDia1 (*Figure 3D,E*). The fractions of the time when FHT occupied >1% of the volume fraction of A1 in the two replicates (350 ns for the first replicate and 200 ns for the second replicate) were, respectively, 64% and 21% for Cdc12, 19% and 18% for Bni1 and 4% and 2% for mDia1. Thus, both FHL and FHT of Cdc12 created more steric interference for subunit addition than Bni1 or mDia1.

## How do the three formin FH2 domains influence the helical twist of the barbed end?

We tested the hypothesis that gating is caused by FH2 domains flattening the helical twist of the barbed end of the filament (*Paul et al., 2008*; *Paul and Pollard, 2009b*) with three simulation experiments: (1) extended AA simulations of seven-mer filaments (representing the state following subunit addition); and (2) AA simulations and (3) metadynamics (*Laio and Parrinello, 2002*; *Dama et al., 2015*) simulations of the five-mer filaments (representing the state prior to subunit addition).

### AA simulations of 7-mers

During the first 100 ns of the simulations with each of the formins, the root mean-squared deviations (RMSD) of the alpha-carbon atoms changed before stabilizing (*Figure 2D*). The Bni1 FH2-actin complex changed the most during this time (*Figure 2D*, from Baker et al. [*Baker et al., 2015*]), since the model actin filament started with a 167° twist and the FH2 contacts came from a co-crystal structure with a 180° twist of the actin helix. In the Baker simulations of Bni1 FH2 domains (*Baker et al., 2015*), the twist angle relaxed from 167° to ~172°, the conformation used here for the homology models of Cdc12 and mDia1 FH2 domains. The RMSDs of these filaments with Cdc12 and mDia1 FH2 domains changed less than Bni1 during the first 100 ns of simulations (*Figure 2D*).

During the subsequent 400 ns of AA MD simulations, the actin filament seven-mer with mDia1 FH2 relaxed toward twist angles more favorable for binding an incoming actin subunit than filaments with Bni1 or Cdc12 (*Figure 4*). The twist angles between the first two pairs of subunits of all three samples decreased continuously from ~172° (*Figure 4D,F*) as they fluctuated on shorter time scales (*Figure 4A–C*). The angles between the first and second pairs of actin subunits were strongly correlated. After 500 ns of simulations, the twist angles of the terminal actin subunits associated with the mDia1 FH2 ($twist_{A1-A2} = 169.4° \pm 0.7°$ and $twist_{A2-A3} = 167.2° \pm 1.1°$, means ± SD) were close to 167° while those associated with Cdc12 ($twist_{A1-A2} = 171.3° \pm 0.4°$ and $twist_{A2-A3} = 169.6° \pm 0.5°$) or Bni1 ($twist_{A1-A2} = 170.7° \pm 0.4°$ and $twist_{A2-A3} = 170.2° \pm 0.5°$) FH2 fluctuated mostly in the angle range of 169–173° (*Figure 4E,G*). The angles between subunits further from the barbed end were more

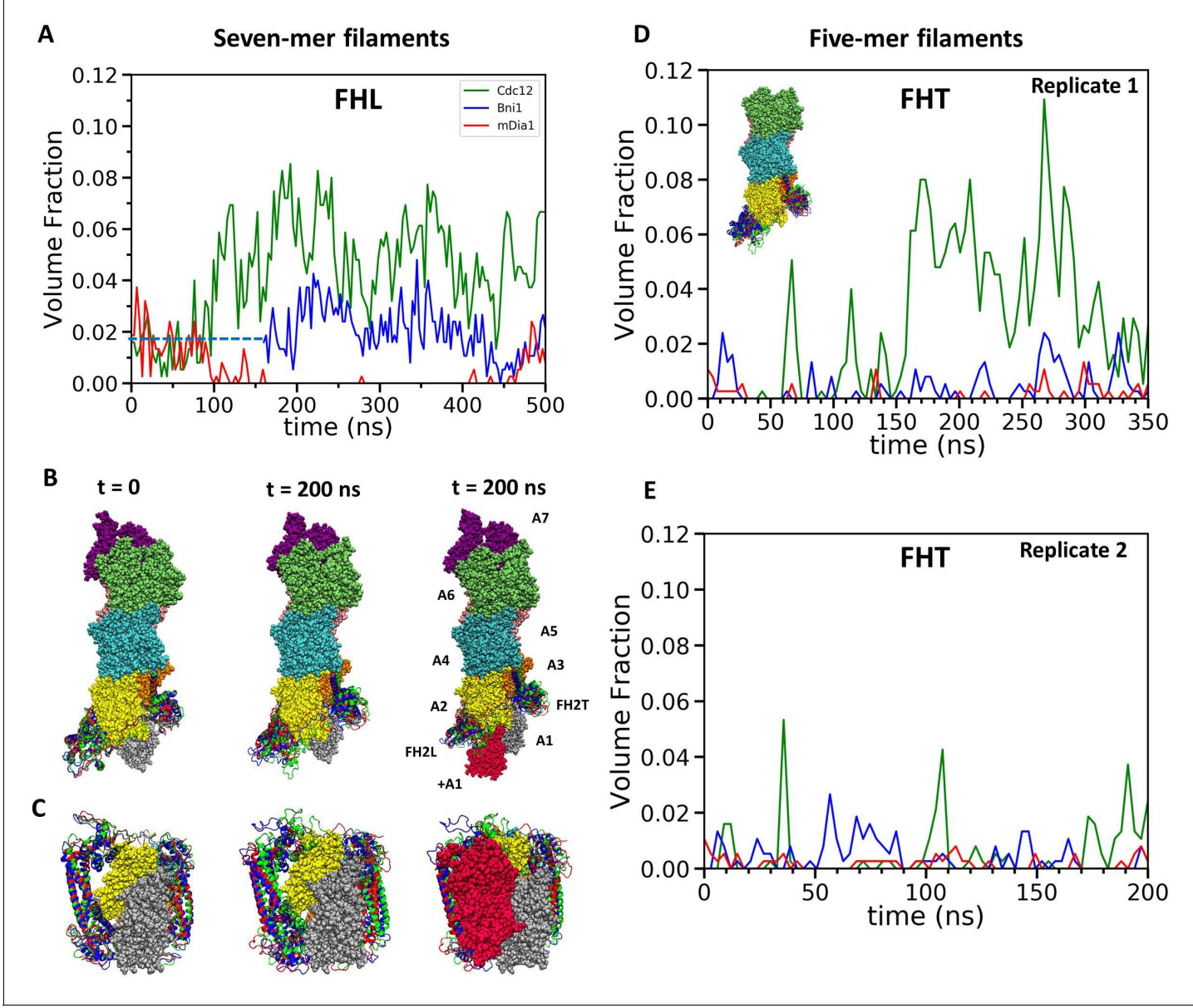

**Figure 3.** Steric clashes of FH2 dimers with incoming actin subunits during all-atom simulations of seven-mer and five-mer filaments. (A–C) FH2 dimers on seven-mer filaments including actin A1. (A) Time course of volume fractions of actin subunit (+A1) occupied by FHL during 500 ns simulations. Measurements start at 0 ns for Cdc12 and mDia1, and at 160 ns for Bni1. (B, C) Ribbon diagrams of Cdc12, Bni1 and mDia1 FH2 domains interacting with the barbed end of a space-filling model of an actin filament at the beginning (left) and at the 200 ns (middle) MD simulations. In the models on the right, red actin subunit +A1 was added at the end of the simulations. (B) Side views. (C) Views from the barbed end. (D, E) FH2 dimers on five-mer filaments without actin subunits A1 and A7. Time courses of volume fractions of incoming actin subunit (A1) occupied by FHT during (D) 350 ns simulations for the first replicas and (E) 200 ns simulations for the second replicas. The initial structures were generated by removing subunits A1 and A7 at the end of 200 ns all-atom simulations of seven-mer filaments from (A–C). FH2 dimers on a five-mer filament without actins A1 and A7 is shown in the inset.

DOI: https://doi.org/10.7554/eLife.37342.010

variable during the 500 ns AA simulations (*Figure 4—figure supplement 1*). The pointed end subunits (twist angles between A6-A7 and A5-A6) were close to 167° on average as they are in the middle of long actin filaments, but they varied more over time due to the presence of fewer neighboring subunits. Therefore, the FH2 domains of Cdc12 and Bni1 only flattened the end of the filament locally.

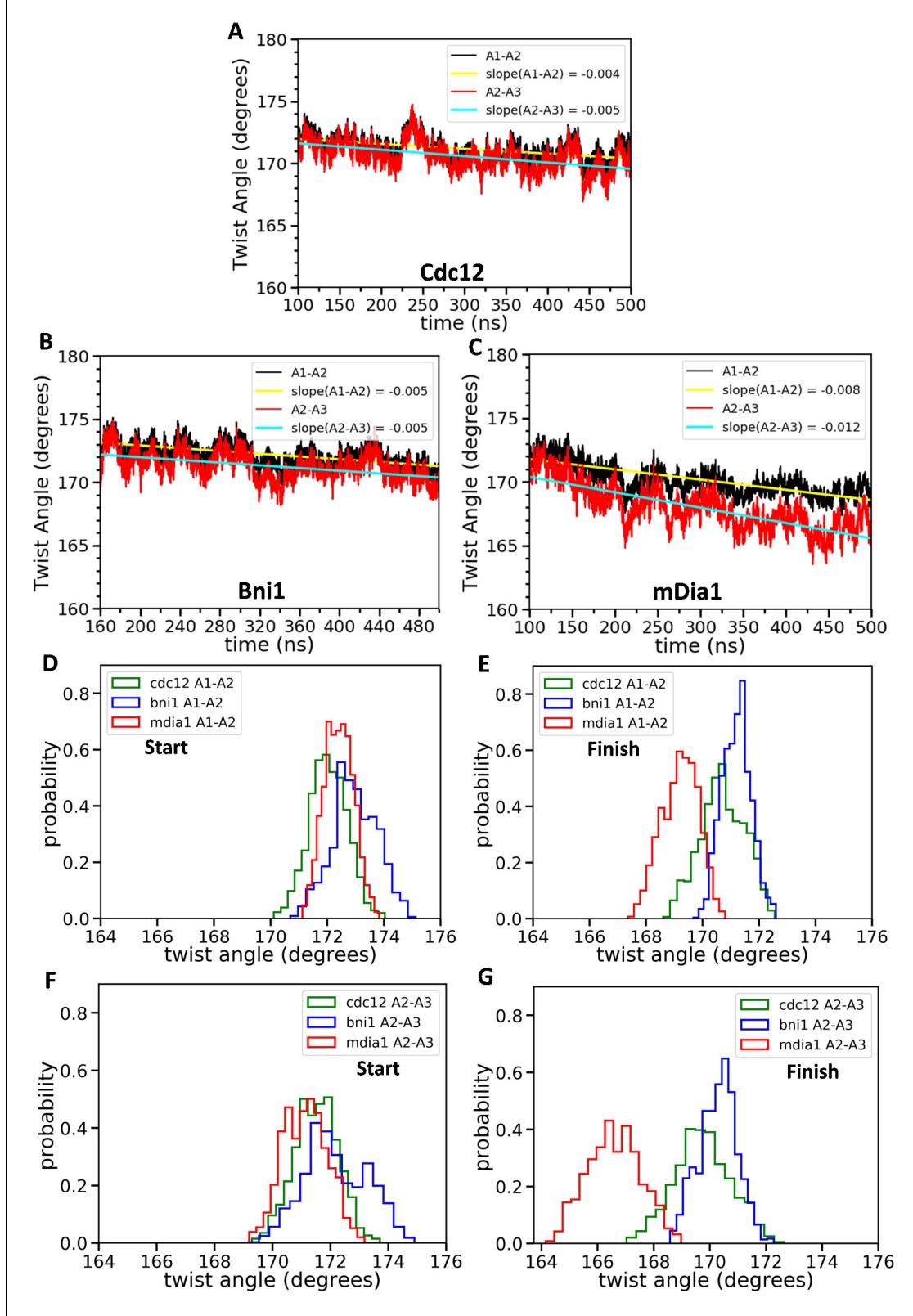

**Figure 4.** Effect of FH2 domains on the barbed end configurations of actin seven-mer filaments. The panels show helical twist angles between subunits at the barbed ends of actin filaments associated with three different formin FH2 dimers during all-atom MD simulations. (**A–C**) The angles between the actin subunits (**A1–A2 and A2–A3**) as a function of time. The measurements start after initial equilibrations (*Figure 2*) at 100 ns for Cdc12 and mDia1 or

*Figure 4 continued*

at 160 ns for Bni1. (D–G) Comparison of the distributions of angles between actin subunits A1-A2 and A2-A3 during two different time intervals. (D, F) t = 100–150 ns for Cdc12 and mDia1, and t = 160–210 ns for Bni1. (E, G) t = 450–500 ns for all systems.

DOI: https://doi.org/10.7554/eLife.37342.011

The following source data and figure supplement are available for figure 4:

**Source data 1.** Twist angles between actin subunits as a function of time.

DOI: https://doi.org/10.7554/eLife.37342.013

**Figure supplement 1.** Effect of FH2 domains on the configurations of actin seven-mer filaments.

DOI: https://doi.org/10.7554/eLife.37342.012

## AA simulations of 5-mers

Two separate AA simulations of five-mer filaments revealed that each of the three formin FH2 domains favors distinctly different twist angles between subunits A2 and A3, the two subunits that form the binding site for incoming subunit A1 (*Figure 5A*). The most probable twist angles in the two replicates were 168.2° ± 0.5° and 170.3° ± 0.7° for mDia1, 172.2° ± 0.8° and 171.6° ± 1.2° for Bni1 and 173.9° ± 0.8° and 175.1° ± 0.7° for Cdc12 (*Figure 5B,C*) with fluctuations for each over a range of about 4°. The maximum difference in the twist angles at the actin barbed ends interacting with Cdc12 and mDia1 reached up to 9° during these simulations. Therefore, mDia1 FH2 favored the 'open' configuration of barbed ends awaiting the association of a subunit, while Cdc12 favored twist angles that compromise subunit addition.

## MB-MetaD simulations of 5-mers

We used a new metadynamics enhanced free-energy sampling method called Metabasin Metadynamics (MBMetaD) (*Dama et al., 2015*) for two independent simulations of five-mer filaments to investigate the mobility of FH2 domains on barbed ends and the influence of FH2 domains on the twist angle between subunits A2 and A3. The five-mer filaments were generated by removing actin subunit A1 from the barbed end and A7 from the pointed end of the seven-mer filament at the end of the first 200 ns of AA simulations (*Figure 3D*, inset). MB-MetaD adds time-dependent Gaussian 'hill' potentials to the energy landscape to discourage the system from visiting already-sampled states and this enabled FH2 domains to escape from their initial conformations. This simulation approach enhances sampling by increasing transition rates between positions in a system's free-energy landscape, making it feasible to investigate physical phenomena inaccessible by standard AA MD. These energy states are of interest, because strong intermolecular interactions may create energy barriers that trap FH2 domains in 'closed' or 'open' states on the filament. Thus, MetaD simulations tested the reliability of the simulations of the homology models by sampling a wider range of conformations independent of their initial configurations.

The 80 ns MB-MetaD simulations confirmed that mDia1 FH2 favored 'open' twist angles near 168°, while the Cdc12 FH2 favored the 'closed' twist angles > 170° (*Figure 6C*). The Bni1 FH2 favored intermediate twist angles but the distribution resembled mDia1 more than Cdc12.

## How do interactions of FH2 domains with barbed ends explain the range of gating factors?

We used the results of our simulations to examine in detail how each formin interacts with actin subunits at the barbed end of a filament. We considered buried surface area, contacts and salt bridges.

### Buried surface area

During the last 20 ns of the 200 ns AA simulations, contacts between the FH2 domains and actin filament buried the following total surface areas: Cdc12 FHL 5493 ± 341 Å$^2$ (mean ± SD), Cdc12 FHT 5232 ± 207 Å$^2$, Cdc12 total 10,725 Å$^2$; Bni1 FHL 4752 ± 205 Å$^2$, Bni1 FHT 5364 ± 247 Å$^2$, Bni1 total 10,116 Å$^2$; and mDia1 FHL 4938 ± 212 Å$^2$, mDia1 FHT 5635 ± 250 Å$^2$, mDia1 total 10,573 Å$^2$. Thus, the differences in buried surface area do not correlate with the gating factors.

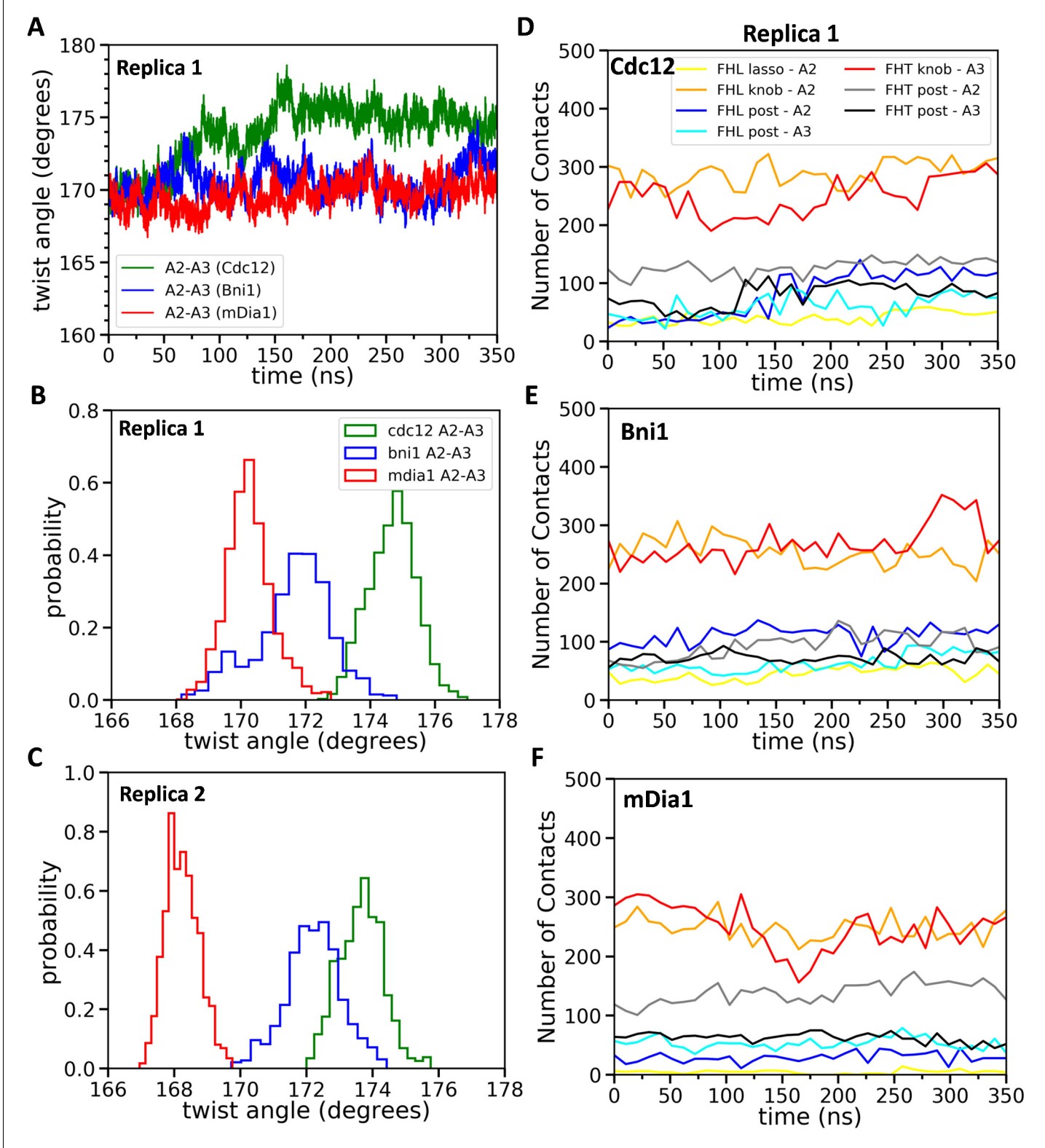

**Figure 5.** Effect of FH2 domains on the barbed end configurations of actin five-mer filaments before the addition of actin subunit A1. (**A**) The helical twist angles between actin subunits A2 and A3 as a function of time during 350 ns of all-atom MD simulations (replica 1) of the FH2 domains of Bni1, Cdc12 and mDia1 associated with the actin filament five-mer consisting of subunits A2, A3, A4, A5 and A6. The initial structures in these simulations were generated by removing subunits A1 and A7 at the end of 200 ns all-atom simulations of the FH2 domains of Bni1, Cdc12 and mDia1 associated with the actin filament seven-mer. (**B, C**) Comparison of the distributions of angles between actin subunits A2 and A3 during the last time intervals of

*Figure 5 continued on next page*

*Figure 5 continued*

two independent simulations (replica 1 and replica 2). (B) The last 50 ns of the 350 ns simulations. (C) The last 20 ns of the 200 ns simulations. (D–F) The number of contacts between the lasso, knob and post regions of the FH2 domains and actin subunits A2 and A3 as a function of time during 350 ns of the all-atom MD simulations (replica 1).

DOI: https://doi.org/10.7554/eLife.37342.014

The following source data and figure supplements are available for figure 5:

**Source data 1.** Twist angles between actin subunits as a function of time.
DOI: https://doi.org/10.7554/eLife.37342.017
**Figure supplement 1.** The interactions of FH2 domains (post regions) with an actin filament and the barbed end configuration.
DOI: https://doi.org/10.7554/eLife.37342.015
**Figure supplement 2.** The interactions of FH2 domains (lasso and knob regions) with an actin filament and the barbed end configuration.
DOI: https://doi.org/10.7554/eLife.37342.016

## Contacts

We measured the contacts between all residues of the three FH2 domains and both five-mer (*Figure 5D–F*) and seven-mer (*Figure 7A–F*) actin filaments as well as the total nonbonded interaction energies (sum of van der Waals and electrostatic interactions) of these contacts (*Figure 7G–I*). Owing to the symmetry of the FH2 dimers, the contacts of FHL with A1 and A2 resembled the contacts of FHT with A2 and A3. FHT made more contacts than FHL, because of the presence of the terminal subunit A1 in seven-mer filaments. The lasso, linker, knob and post regions of both FH2 domains of all three formins contributed to these contacts with the actin filament, with major contacts between FHT and A1 and A3 and between FHL and A2. Mutations of formin FMNL3 had implicated the post and lasso regions in gating: mutations K800A and R782A in the post and R570A in the lasso slowed actin filament elongation, presumably by decreasing the gating factor of ~0.4 (*Thompson et al., 2013*). Although the knob regions are the most variable part of the FH2 domains, they made the most extensive contacts with actin filaments, especially with the barbed end grooves of subunits A2 (by FHL) and A3 (by FHT).

During AA simulations of five-mer filaments, flattening of the barbed end by Cdc12 FH2 was strongly correlated with the number of contacts between actin subunits A2 and A3 and the post regions of both FHT and FHL (*Figure 5—figure supplement 1* and *Supplementary file 3*). The sharp increase in the number of contacts for twist angles > 173° was also seen as an abrupt change in the time course data in *Figure 5D* (blue line is contacts between FHL post and A2). This correlation does not establish causality, but the additional contacts may provide the free-energy change for the unfavorable change in the conformation of the filament. On the other hand, the contacts between the Cdc12 FH2 lasso or knob regions of Cdc12 and actin subunits A2 and A3 did not change as the filament flattened (*Figure 5—figure supplement 2*). During these simulations, neither Bni1 nor mDia1 caused a large change in the twist angles or the numbers of contacts of the knob, lasso or post regions with actin (*Figure 5—figure supplement 2*, *Figure 5—figure supplement 1* and *Supplementary file 3*).

Other than this behavior of Cdc12, extensive analysis of the contacts and interaction energies between the FH2 domains and actin filaments (*Figures 5* and *7*, *Figure 5—figure supplement 2* and *Figure 5—figure supplement 1* and *Supplementary file 3* and *Supplementary file 4*) did not show a simple, consistent correlation of the numbers of contacts or interaction energies with gating factors. Several measurements were not correlated with gating factors: the total interaction energy of Cdc12 was the largest (−713 kcal/mole), but interactions were stronger with mDia1 (−396 kcal/mole) than Bni1 (−204 kcal/mole); the lasso and post regions of mDia1 made fewer contacts with actin than Cdc12 and Bni1 (*Figure 7—figure supplement 1*), while the linkers of mDia1 made more contacts than Cdc12 and Bni1; and the knob region of Bni1 FHT had fewer contacts with actin subunit A3 than Cdc12 and mDia1 FHT but more contacts with actin subunit A1.

## Salt bridges

The AA simulations of both five-mer and seven-mer filaments revealed that the number of salt-bridges between the FH2 knobs and the barbed end groove of actin correlate with gating (*Table 1*). At two time points in the simulations of seven-mer filaments and at the end of the simulation of five-

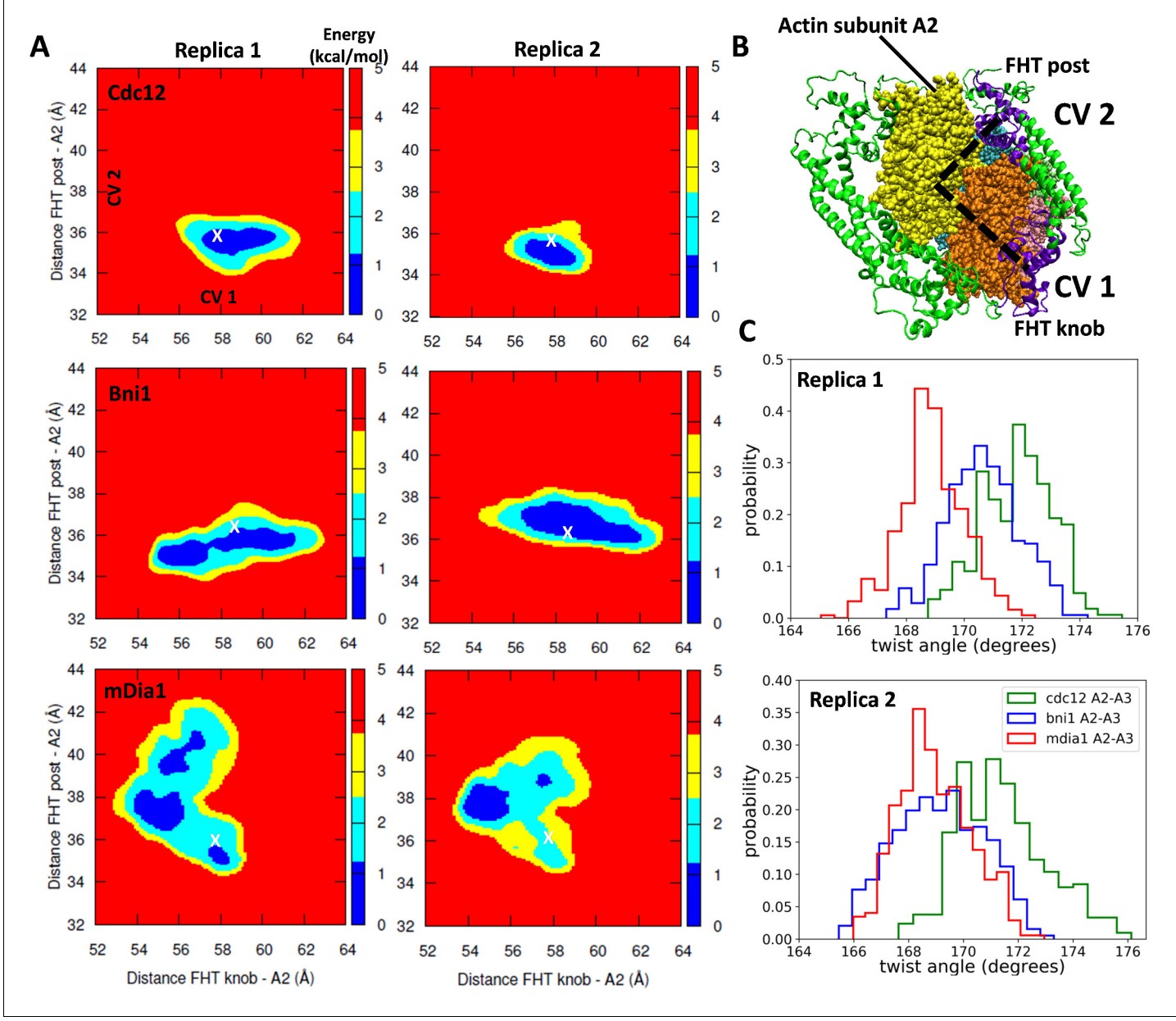

**Figure 6.** Free-energy sampling of the conformational mobility of FHT domains interacting with five-mer filaments. (**A**) Two independent metabasin metadynamics (MBMetaD) simulations for each formin were carried out (run for 80 ns) to understand the conformational mobility of the FHT domain. The collective variables (CVs) defined in (**B**) were selected to describe the mobility of the FHT domain in the region of the incoming actin subunit A1 (shown by dashed black lines). The first CV (CV1) is the distance between the center of mass (COM) of FHT knob and the COM of actin subunit A2. CV2 is the distance between the COM of the FHT post and the COM of actin subunit A2. Initial distances from the knob and post to the COM of A2 are marked with 'X'. If a larger area was favored to be explored, that would mean that the incoming actin monomer could be more easily accommodated into the barbed end as the rearrangement of the FHT domain requires a lower energy barrier. (**B**) Space-filling models of the barbed end of the filament (subunit A2 is yellow and subunit A3 is brown) and ribbon diagrams of the FH2 domains of Cdc12 (green with the knob and post of FHT in purple). (**C**) The distributions of angles between actin subunits A2 and A3 during two independent MBMetaD simulations.
DOI: https://doi.org/10.7554/eLife.37342.018

mer filaments Cdc12 had more salt bridges occupied for large fractions of time than Bni1 or mDia1. Note that these salt bridges were dynamic. For example, the Cdc12 FH2 knob helices formed two additional salt bridges and the residues making some salt bridges rearranged between the middles and ends of the 500 ns AA MD simulations. The salt bridges varied more during simulations of the

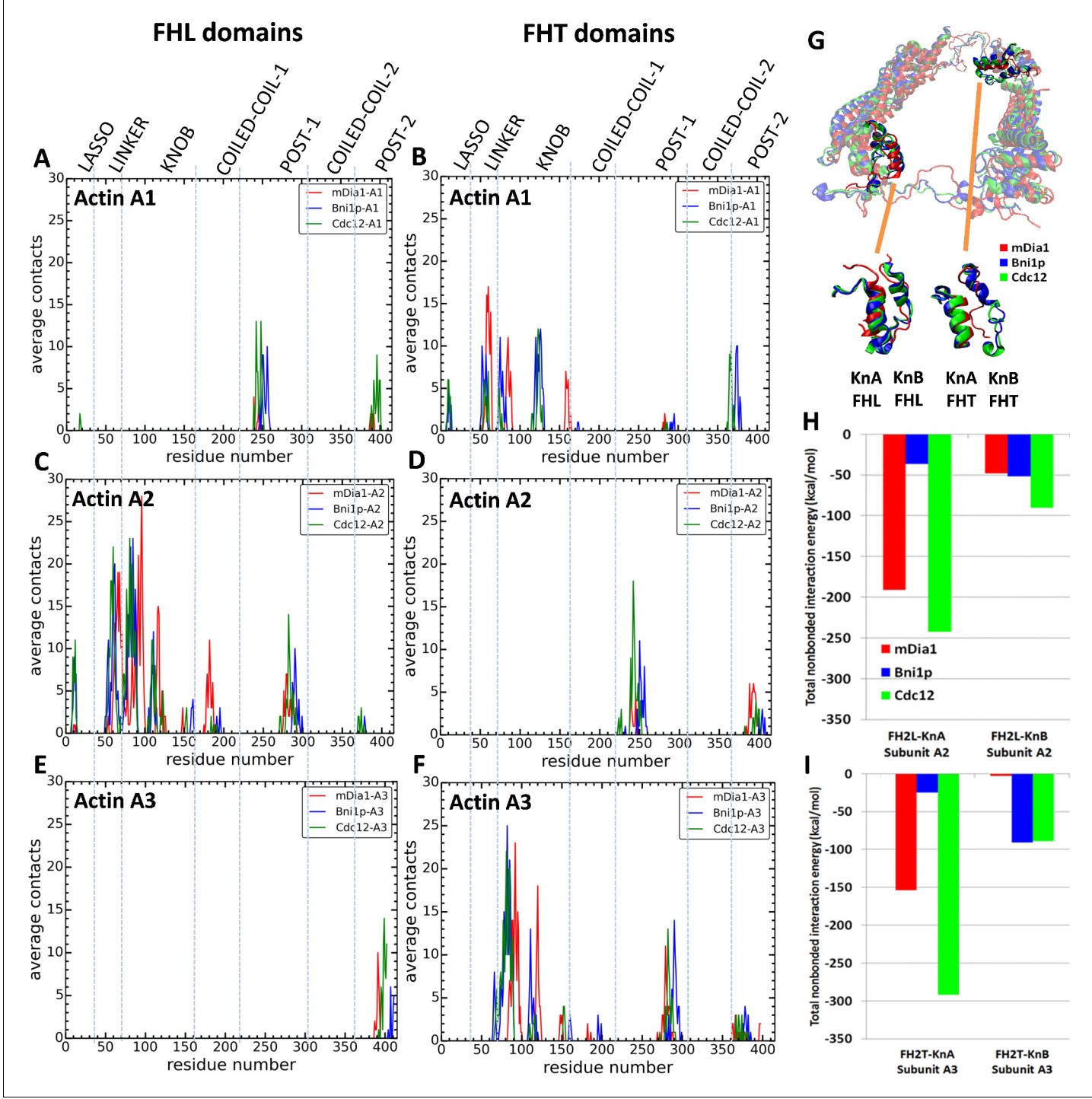

**Figure 7.** Interactions of FH2 domains with actin filament seven-mers. The time-averaged number of contacts over the last 20 ns of the 200 ns simulations between the three actin subunits at the barbed ends of the filaments and the FH2 domains of mDia1 (red), Bni1 (blue) and Cdc12 (green). The actin subunits are numbered from 1 to 3, starting with the newly added A1 subunit. (A, B) A1 actin subunits; (C, D) A2 actin subunits; and (E, F) A3 actin subunits. (A, C, E) show the contacts of the FHL domains and (B, D, F) show the contacts of the FHT domains. A pair of residues was considered to be in contact if the distance between their C-alpha atoms was ≤12 Å. (G) View from the pointed end of the structural alignment of FH2 domains taken from the end of 200 ns all-atom simulations of Bni1, Cdc12 and mDia1 FH2 domains interacting with an actin filament seven-mer. The orange lines point to the knA and knB helices of mDia1, Cdc12 and Bni1 FH2 domains. (H, I) Total nonbonded interaction energy (sum of van der Waals and electrostatic interactions) between (H) FHL (leading) and (I) FHT (trailing) knob helices of mDia1, Bni1, Cdc12 formins and the barbed end of actin subunits A2 and A3.

*Figure 7 continued on next page*

*Figure 7 continued*

DOI: https://doi.org/10.7554/eLife.37342.019
The following figure supplement is available for figure 7:

**Figure supplement 1.** Interactions of each region in FH2 domains with an actin filament. The number of contacts between the lasso, linker, knob and post regions of the FH2 domains and actin subunits (**A1, A2 and A3**) as a function of time during last 50 ns of 500 ns all-atom MD simulations.
DOI: https://doi.org/10.7554/eLife.37342.020

five-mer filaments (*Table 1*), but Cdc12 again formed more (nine) salt bridges with the actin barbed end groove than Bni1 (two) or mDia1 (one).

During AA MD simulations of the seven-mer actin filaments the lasso/linker regions of the mDia1 FH2 domains formed more salt bridges with actins A1, A2 and A3, but both Cdc12 and Bni1 formed more long-lived salt bridges (i.e. occupied more than 70% of the time) (*Table 2*). Only aspartic acid (D363) in the A2 subunit of the actin filament formed a salt bridge with the lasso/linker regions of all three formins (*Table 2*). It was occupied 100% of the time in Cdc12, 85% in Bni1 and only 23% in mDia1.

**Table 1.** Stability of salt bridges between formin FH2 domains and actin subunits.
Stability was measured as the percent of the time that a salt bridge formed between knob helices of the three FH2 domains and actin barbed end grooves of actin subunits A2 or A3 during (A) the last 20 ns of AA simulations spanning 200 ns of seven-mer filaments, (B) the last 50 ns of the simulations spanning 500 ns of seven-mer filaments and (C) the last 50 ns of the simulations spanning 350 ns of five-mer filaments.

**Cdc12 knob helices/Actin barbed end groove**

| FH2 residue | FH2 domain | Actin residue | Actin subunit | (A) Percent formed | (B) Percent formed | (C) Percent formed |
|---|---|---|---|---|---|---|
| K1068 | FHL | D25 | A2 | 78.2 | 76.9 | 86.5 |
| E1090 | FHL | R147 | A2 | - | 63.1 | 13.1 |
| E1093 | FHL | R147 | A2 | 99.4 | - | - |
| K1099 | FHL | E167 | A2 | - | - | 41.2 |
| K1105 | FHL | E167 | A2 | - | 52.6 | 77.1 |
| K1107 | FHL | E167 | A2 | 41.5 | - | - |
| K1068 | FHT | D25 | A3 | 95.2 | 70.7 | 82.4 |
| K1072 | FHT | D25 | A3 | 64.3 | 71.4 | 85.3 |
| K1072 | FHT | D24 | A3 | - | 15.6 | 12.8 |
| E1093 | FHT | R147 | A3 | 94.8 | 97.3 | 99.8 |
| K1099 | FHT | E167 | A3 | - | 47.2 | 18.1 |

**Bni1 knob helices/Actin barbed end groove**

| | | | | | | |
|---|---|---|---|---|---|---|
| E1463 | FHL | R147 | A2 | 83.6 | 89.5 | 88.6 |
| R1423 | FHL | E167 | A2 | 0.401 | 0.541 | - |
| E1463 | FHT | R147 | A3 | 90.0 | 88.5 | 93.8 |
| K1467 | FHT | E167 | A3 | 97.4 | 95.7 | - |

**mDia1 knob helices/Actin barbed end groove**

| | | | | | | |
|---|---|---|---|---|---|---|
| R851 | FHL | D25 | A2 | 97.8 | 93.5 | - |
| E871 | FHL | R147 | A2 | 8.42 | 18.8 | - |
| K879 | FHL | E167 | A2 | 25.3 | 3.69 | - |
| K838 | FHT | E167 | A3 | 98.6 | 98.6 | 25.7 |

DOI: https://doi.org/10.7554/eLife.37342.021

**Table 2.** Salt bridges between formin FH2 domains and actin subunits.

Percentage of the time that salt bridges formed between the lasso and linker regions of Cdc12, Bni1 and mDia1 FH2 domains and actin subunits during the last 20 ns of AA simulations spanning 200 ns of seven-mer filaments.

| FH2 residue | FH2 domain | Actin residue | Actin subunit | % Formed |
|---|---|---|---|---|
| Cdc12/actin | | | | |
| R990 | FHL-lasso | D363 | A2 | 100. |
| K992 | FHT-lasso | E125 | A1 | 99.6 |
| K1038 | FHL-linker | E99 | A2 | 71.5 |
| K1041 | FHL-linker | E100 | A2 | 52.5 |
| K1045 | FHL-linker | E2 | A2 | 72.3 |
| K1045 | FHL-linker | D3 | A2 | 93.2 |
| K1046 | FHL-linker | E4 | A2 | 50.3 |
| K1038 | FHT-linker | D363 | A1 | 20.6 |
| Bni1/actin | | | | |
| K1357 | FHL-lasso | D363 | A2 | 85.2 |
| K1357 | FHT-lasso | D363 | A1 | 23.0 |
| K1359 | FHT-lasso | E125 | A1 | 99.2 |
| R1402 | FHL-linker | E99 | A2 | 99.4 |
| E1403 | FHL-linker | K359 | A2 | 95.0 |
| K1410 | FHL-linker | E4 | A2 | 24.0 |
| K1410 | FHL-linker | E100 | A2 | 81.8 |
| K1412 | FHL-linker | D3 | A2 | 57.5 |
| R1402 | FHT-linker | D363 | A1 | 98.4 |
| mDia1/actin | | | | |
| R764 | FHL-lasso | D363 | A2 | 23.4 |
| K807 | FHL-linker | E125 | A2 | 49.9 |
| K813 | FHL-linker | E4 | A2 | 37.9 |
| K813 | FHL-linker | E99 | A2 | 20.4 |
| K826 | FHL-linker | D3 | A2 | 70.9 |
| K826 | FHL-linker | E2 | A2 | 43.3 |
| K827 | FHL-linker | E2 | A2 | 13.2 |
| K828 | FHL-linker | E2 | A2 | 11.4 |
| K807 | FHT-linker | E125 | A1 | 23.0 |
| K810 | FHT-linker | E83 | A1 | 43.7 |
| K810 | FHT-linker | E125 | A1 | 46.7 |
| K813 | FHT-linker | D51 | A1 | 99.0 |
| E816 | FHT-linker | R37 | A1 | 96.8 |
| K828 | FHT-linker | E2 | A3 | 25.1 |

DOI: https://doi.org/10.7554/eLife.37342.022

## Energy barriers

The MB-MetaD simulations revealed the free-energy barriers to motions of FH2 domains on the ends of filaments (*Figure 6A*). At the outset of the MB-MetaD simulations the steric clashes between the FHT domains of the three formins and the incoming actin subunit A1 differed (see *Figure 3D,E*), with the approximate initial distances from the knob and post to the center of mass of A2 marked with X in *Figure 6A*. The FHT domains must rearrange to escape from these steric clashes. The energy landscapes in *Figure 6A* showed that the FHT domain of Cdc12 was more confined by high-

energy barriers (yellow and red regions) than Bni1 or mDia1. These high-energy barriers for conformational rearrangements restricted the FHT domain of Cdc12 to a smaller area of the conformational space (including those with steric clashes with A1) than the FHT domains of Bni1 and mDia1. The lower conformational mobility of the Cdc12 FH2 domain may be related to large numbers of salt bridges (*Table 1*) and correlates with a larger distortion of the barbed end (*Figure 6A,C*). Since the strength of the interactions between FH2 domains and actin play a role in actin filament nucleation (*Baker et al., 2015*), Cdc12 may be the best nucleator of these three formins.

## Coarse-grained simulations over longer time scales

We used simulations of coarse-grained (CG) models of FH2-actin structures to study deviations from their initial configurations at large spatiotemporal scales beyond the range of AA MD simulations. CG MD simulations can reveal essential intra- and inter-molecular interactions by approximating other non-essential interactions. CG simulations also allowed us to test the importance of electrostatic interactions at the FH2-bound barbed ends by varying the dielectric constant at the protein surfaces. The dielectric constant depends on the ionic strength in a complex way (*Levy et al., 2012*), and there is no consensus on the effective dielectric constant at the protein-protein interface despite numerous investigations (*Schutz and Warshel, 2001*; *Li et al., 2013*). Therefore, the dielectric constant at the protein-protein interface was varied to tune the strength of charge-charge interactions and investigate its effect on the behaviors of formin-actin complexes. Increasing the dielectric constant will reduce the effective electrostatic interactions between the formin and actin filament in the CG model, so this variable can be used to assess the role of electrostatic interactions in the conformations of formin FH2 domains at the barbed end of an actin filament.

We created CG models of actin–FH2 domain structures for Cdc12, Bni1 and mDia1 using essential dynamics coarse graining (EDCG) (*Zhang et al., 2008*). Hetero elastic network modeling (hetero-ENM) (*Lyman et al., 2008*) was used for the intramolecular interactions of the proteins, and the intermolecular interactions between the formins and the actin filaments were described by a combination of van der Waals and electrostatic interactions using Lennard Jones and Screened Coulomb (Yukawa) potentials (see Materials and methods section for the details). For example, the CG model of the Cdc12 FH2 domain consisted of 45 CG sites assigned by the ED-CG method (*Figure 8A*) that were connected by heteroENM effective harmonic interactions (*Figure 8B*) to match the RMSF of all-atom and CG simulations (*Figure 8D*). We combined CG models of FH2 and actin to build CG models of the FH2–actin complex (*Figure 8C*) and ran 2 μs simulations of seven-mer filaments with dielectric constants (ε) of 1 and 3. (It should be noted that CG MD time is of a significantly larger scale factor than AA time.)

The FH2 domains of the three formins each responded differently during multiple CG simulations of the seven-mer actin filament model. At low dielectric constant (ε = 1), the three formins deviated only slightly from their intial configurations (*Figure 8E*). Over the course of CG simulations with a higher dielectric constant (ε = 3) the Cdc12 FH2 domains underwent larger fluctuations than the Bni1 and mDia1 FH2 domains (*Figure 8F*). The behaviors of Cdc12 and mDia1 were more clearly separated from each other than from Bni1 in the AA MD simulations, where the behavior of Bni1 mostly varied between those of Cdc12 and mDia1. As the parameterization of the CG model was based on the AA data, Bni1 demonstrated a similar behavior in the CG simulations by switching between two conformations (the conformation of Bni1 can become more similar to either the conformation of Cdc12 or that of mDia1 (*Figure 8F*)). This is also consistent with experimental findings as the gating factor of Bni1 (~0.5–0.7) reflects its ability to switch between open and closed states. The striking impact of the dielectric constant on the CG simulations of Cdc12 are consistent with our AA simulations showing stronger electrostatic interactions between Cdc12 and the actin filament than Bni1 and mDia1.

Although barbed ends with FH2 domains simply fluctuated around their equilibrium configurations without flattening during the CG simulations (owing to the effective harmonic bonds between the subunits in heteroENM), these simulations (*Figure 9*) provided information on steric blocking that complements the AA simulations (*Figure 3*). During the 2 μs CG MD simulations, the center of mass of the Cdc12 FHL domain shifted radially toward the filament axis where it interfered with the incoming actin subunit (*Figure 9C*). The FHL domains of Bni1 and mDia1 fluctuated around their initial positions where intermittant steric blocking was observed in the AA simulations (*Figure 3*).

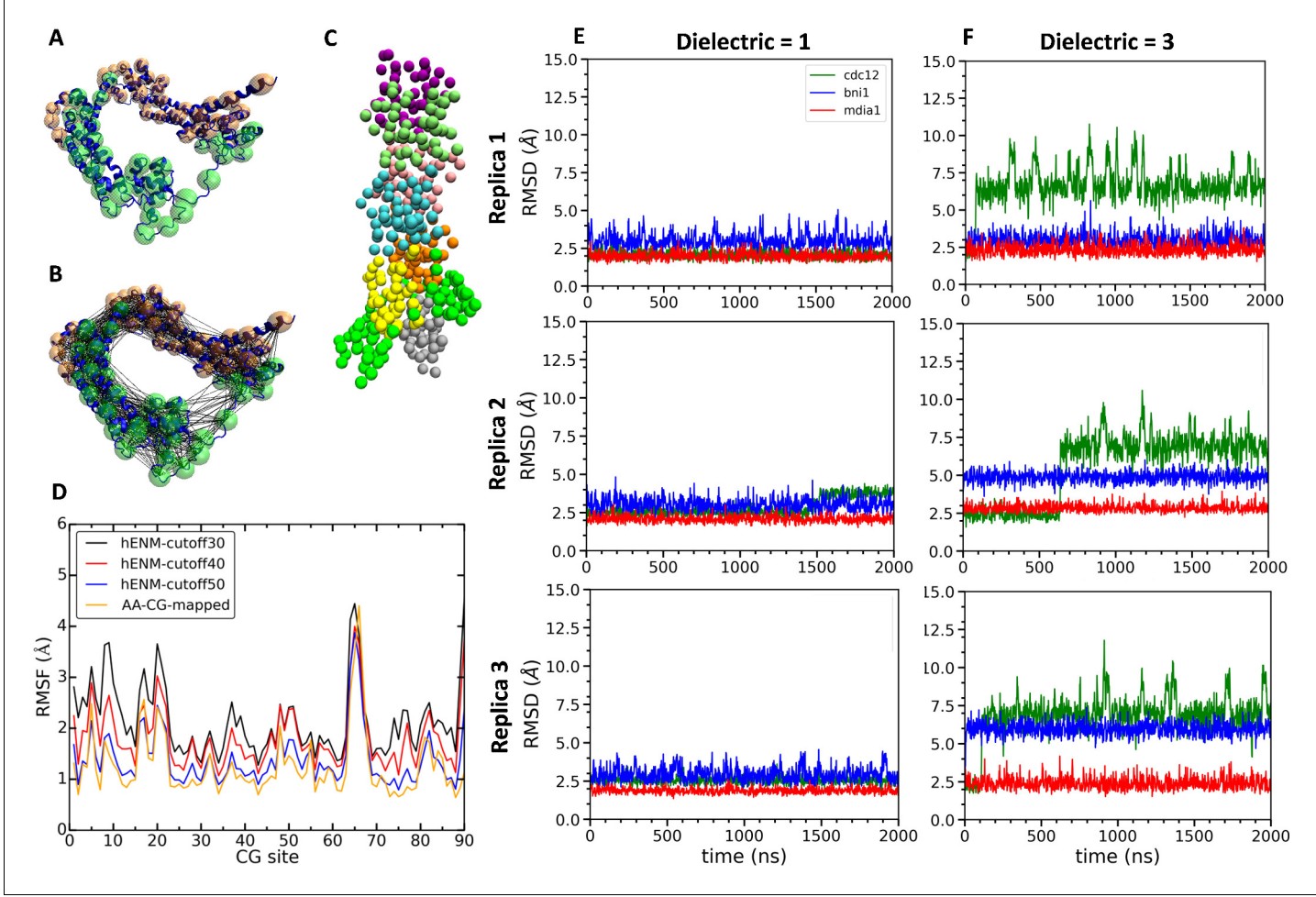

**Figure 8.** Construction and simulations of coarse-grained (CG) models of FH2 domains. (**A**) Assignment of coarse-grained (CG) sites of the Cdc12 FH2 domain dimer by the EDCG method (FHL is green and FHT is orange). (**B**) Connection of CG sites by harmonic bonds defined by heteroENM and matched to the fluctuations observed in all-atom simulations. (**C**) CG model of the Cdc12 FH2 domains on a seven-mer actin filament. FH2 domains are shown in green, actin monomers are colored as follow: A1 (gray), A2 (yellow), A3 (orange), A4 (cyan), A5 (pink), A6 (lime) and A7 (purple). (**D**) Plot comparing root mean square fluctuations (RMSFs) of the CG sites of the Cdc12 FH2 dimer obtained from AA simulations (AA-CG mapped) with RMSFs of these sites in CG simulations using a range of cutoff distances for the harmonic springs that connect every CG site. The cutoff distance of 50 Å (blue) gives the best match with the RMSFs of AA simulations (yellow). (**E, F**) Time courses of the root mean squared displacements of the FH2 dimers from three independent CG simulations spanning two microseconds at low and high dielectric constant. (**E**) Dielectric constant is 1. (**F**) Dielectric constant is 3.

DOI: https://doi.org/10.7554/eLife.37342.023

## Discussion

We used computational modeling to investigate how formin FH2 domains interact with the barbed ends of actin filaments. We used models of three formin FH2 domains with different gating factors to learn how formins slow the elongation of actin filaments. The results of these simulations showed two mechanisms whereby formins interfere with subunit addition at the barbed end, provided information on the equilibrium distribution of open and closed states, and identified the mechanisms responsible for the range of gating factors of the three formins.

### How do formins interfere with subunit addition at the barbed end?

All-atom MD simulations confirmed that FH2 domains on actin filament barbed ends can interfere with the addition of subunits in two ways, both of them scaling with the inhibition of elongation by the three formins. All three formins interfere sterically with the addition of actin subunits by

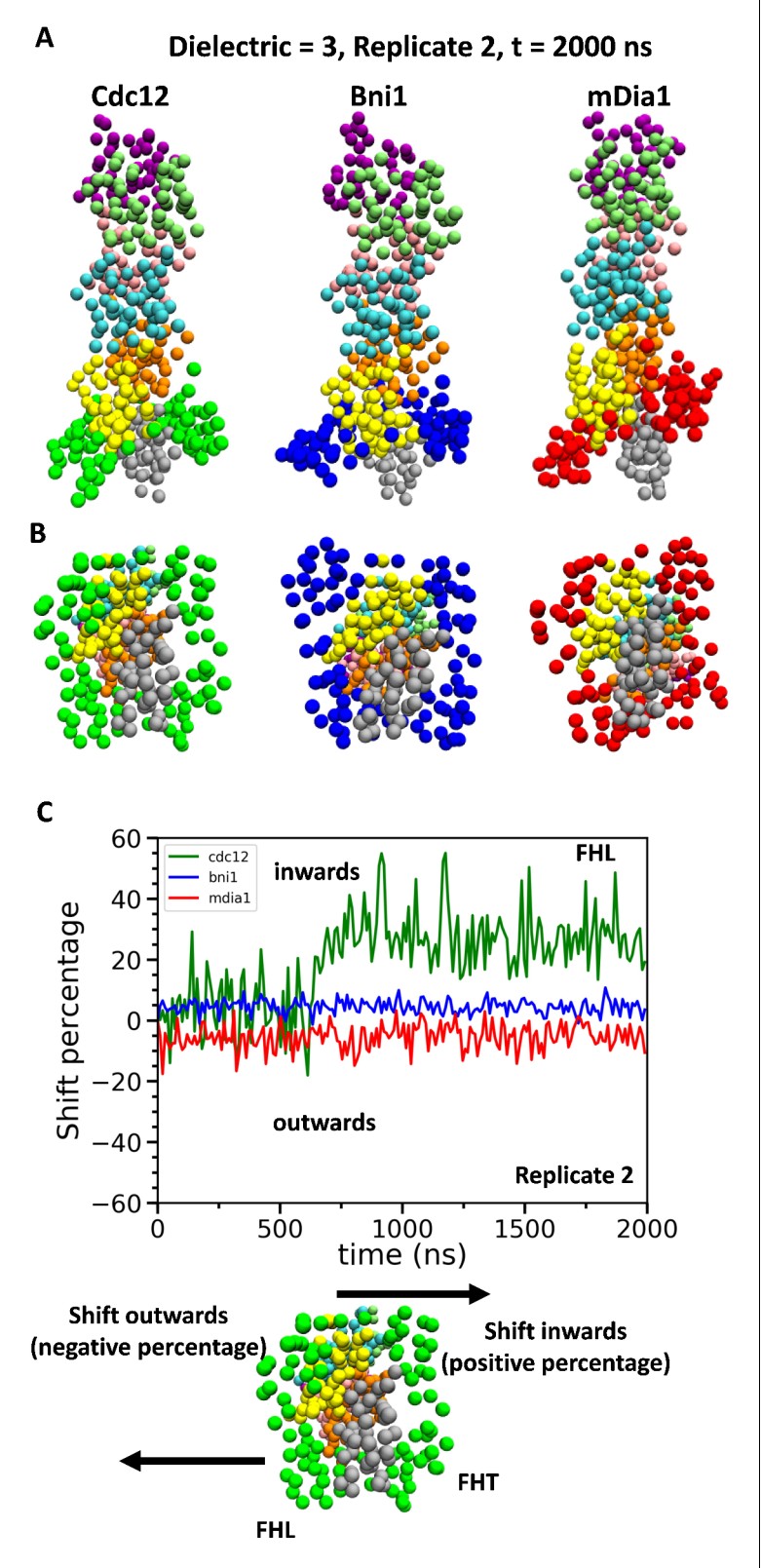

**Figure 9.** Conformational changes of FH2 domains on the barbed ends of actin filament seven-mers during coarse-grained (CG) simulations. (**A**) Side and (**B**) bottom views of structures of Cdc12, Bni1, and mDia1 FH2 domains interacting with the barbed ends of actin filaments at the end (t = 2000 ns) of the CG simulations (dielectric constant of 3, replicate 2). (**C**) Time courses of the center-of-mass displacements of FHL domains from their initial positions (shown as percentage change) in the direction perpendicular to the long axis of the actin filaments. A positive percentage indicates a shift
*Figure 9 continued on next page*

*Figure 9 continued*

towards the actin filament (favors the 'closed' state), whereas a negative percentage indicates a shift away from the actin filament (favors the 'open' state). The shifting of a representative FHL domain in two different directions is schematically shown at the bottom.
DOI: https://doi.org/10.7554/eLife.37342.024

infringing on the volumes occupied by both the A1 and +A1 subunits. All three formin FH2 domains also flatten the helical twist of the barbed ends of both five-mer and seven-mer filaments. The distortion is largest on the terminal subunits in contact with the FH2 domains and dissipates gradually away from the end as considered by *Otomo et al. (2005)* rather than abruptly as originally proposed (*Otomo et al., 2005*; *Goode and Eck, 2007*).

Incoming subunits bind to the two terminal subunits at the barbed end, so a helical twist greater than 167° is unfavorable for elongation, because the elements of the binding site are not aligned. Stair-stepping models (*Otomo et al., 2005*; *Xu et al., 2004*; *Goode and Eck, 2007*) assume that the barbed end subunits remain in the 'strained' ('flattened' in our description) conformation during elongation of the filament. Therefore, these models assume that the incoming actin subunit binds to two subunits with a flattened helical twist. This strained conformation is unlikely to be favorable for elongation.

Coarse-grained simulations support the conclusion that steric interference contributes to gating. At long time scales, the conformations of Cdc12 and mDia1 FH2 domains diverged from each other. The simulations of Cdc12 showed that a partially dissociated FHL domain shifts inwards directly into the binding site for an incoming actin subunit, whereas simulations with mDia1 showed that the partially dissociated FHL shifts outwards away from an incoming subunit. Thus, partial dissociation of some formins can increase steric blocking.

## How do the equilibrium conformations of formins on barbed ends fluctuate in time?

Our simulations demonstrate that stochastic thermal motions allow formin-actin filament complexes to sample a range of conformations on a nanosecond time scale. The probability that a system visits any state depends on the difference in free energy between the equilibrium state and other higher energy states, which are visited less often. Our results show that each of the three formins favors a different equilibrium state for the barbed end; Cdc12 favors larger helical twist angles with more steric blocking than Bni1 and mDia1. The MB-MetaD simulations (*Figure 6A*) illustrate that FH2 domains must overcome energy barriers to deviate from the conformations sampled during AA simulations. On the millisecond time scale relevant to actin filament elongation, barbed ends will visit a wide range of conformations from open 167° twists without steric blocking, to closed 180° twists with steric blocking. However, the distributions will differ such that barbed ends with Cdc12 will be in closed conformations most of the time, while ends with mDia1 will be in open conformations most of the time. Nevertheless, we expect that filaments with any of the formin FH2 domains will reach extreme angles during a fraction of the time, at longer time scales.

Both the helical twist of the terminal subunits and the degree of steric interference vary continuously in time rather than switching between discrete states like ion channels, where the gate is either fully open or closed. Consequently, the probability that a diffusing actin subunit will bind to the barbed end likely declines as function of the steric blocking and deviation of the twist of the terminal subunits from 167°. Therefore, the gating factor is a time-average of the degree to which the formin compromises subunit addition rather than literally being in 'open' or 'closed' states. Normally, the probability that a collision between a diffusing monomer and the barbed end of a filament results in binding is high (~2%) (*Drenckhahn and Pollard, 1986*). Gating should not change the rate constant for collisions with the end of a filament associated with an FH2 domain, but the probability of binding will be less than 2% depending on the particular formin. The rate of elongation of a formin-bound filament is governed by the probability that an incoming actin subunit will find the formin-bound end in a binding-competent, open conformation.

### How is gating related to the continuum of twist and steric interference states?

Both the stair-stepping and stepping second models assume that steric blocking and helix flattening contribute to gating. The two models differ in explaining the transition from a closed state to an open state. The stair-stepping model assumes that partial dissociation of a FH2 domain is necessary to relieve steric blocking while the actin helix remains 'strained' (flattened). The stepping-second model assumes that FH2 domains remain bound to the barbed end subunits as they undergo thermal fluctuations during a rapid equilibrium between open and closed states. Simulations of our five-mer model can help distinguish these models, because the FH2 domains are bound to the same subunits as the Otomo 'blocked, n state' (*Otomo et al., 2005*).

Our simulations revealed that dissociation of an FH2 domain is not required to open the end for elongation. Thermal motions can move bound FH2 domains out of blocking sites and allow addition of a new actin subunit, as assumed by the stepping second model (*Paul et al., 2008*; *Paul and Pollard, 2009b*). Indeed, positioning of FH2 domains at the barbed end without subunit A1 does not bring the helicity of the terminal subunits closer to 167° than when subunit A1 is bound (*Figures 4* and *5*), consistent with a model in which formins impede subunit addition by imposing structural constraints on the actin filament.

Neither our all-atom simulations nor our metadynamics simulations include large scale conformational changes, so they do not rule out the 'stair-stepping' hypothesis that dissociation of one FH2 domain from the end of the filament opens the end of the filament. In fact, our metadynamics simulations revealed energy barriers that might be relevant to the partial dissociation of an FH2 domain from a barbed end of the filament during stepping (*Otomo et al., 2005*). A high-energy barrier indicates that the FHT domain of Cdc12 has a lower mobility than the FHT domains of Bni1 and mDia1.

### Do the effects of force on actin filament elongation by formins inform gating mechanisms?

Appling tension to formins on the barbed end of an actin filament can slow or increase the rate of polymerization. In some cases, force slows elongation: Bni1p FH2 domains without FH1 domains in the absence of profilin (*Courtemanche et al., 2013*); and Cdc12p FH1-FH2 domains with profilin (*Zimmermann et al., 2017*). On the other hand, force can increase the rate of elongation: Bni1p FH1-FH2 domains with profilin (*Courtemanche et al., 2013*); and mDia1 FH1-FH2 domains with profilin (*Jégou et al., 2013*) or without profilin (*Kubota et al., 2017*; *Yu et al., 2017*). Faster elongation by mDia1 FH1-FH2 domains requires that the actin filament is torsionally unconstrained (*Yu et al., 2017*). *Jégou et al. (2013)* proposed an elegant physical model to explain the effects of force on mDia1 based on the assumption that an open state depends on movement of an FH2 domain toward the barbed end of the filament. However, force slows elongation by other formins, so it is premature to generalize or rule out other hypotheses such as force influencing the helical twist and gating of the barbed end (*Courtemanche et al., 2013*).

### Why does Cdc12 slow barbed end elongation more than the other formins?

We compared three formins with different gating factors to identify physical factors that might contribute to their range of gating factors:~0.95 for mDia1, 0.5–0.7 for Bni1 and ~0.05 for Cdc12. Both the degree of steric interference and barbed end flattening by the three formins scale with their ability to inhibit elongation, with Cdc12 having the strongest and mDia1 having the weakest effects, but why is this true?

MB-MetaD simulations showed that energy barriers confine Cdc12 FH2 domains to a narrower range of steric blocking and twist states that interfere with elongation than was observed for Bni1 or mDia1 FH2 domains. Differences in buried surface area, numbers of contacts with actin or total non-bonded interaction energies might contribute to these barriers, but none of these parameters correlated with the gating factors of the three formins except for the association of barbed end flattening with numbers of contacts between the Cdc12 post and actin. On the other hand, CG simulations revealed substantial electrostatic effects that may be related to the numbers, locations and stabilities of salt bridges between FH2 domains and actin, which were identified by the AA simulations. However, the relationship between salt bridges and gating is complex and incompletely understood.

The total numbers of stable salt bridges in the three formin-actin complexes (*Tables 1* and *2*) are correlated with the equilibrium distributions of steric interference and flattening of barbed ends, but the locations of these salt bridges appear to matter. For example, Cdc12 with a small gating factor has more stable salt-bridges between the knob helices and FHL linkers and actin A2 than mDia1 with a large gating factor. However, the opposite is true for salt-bridges between the FHT linkers and actin A1 where mDia1 has a greater number of salt-bridges than Cdc12. Tight binding of FHL to A2 might limit the rearragements associated with FHL shifting from closed states to open conformations without steric interference and with helical twists favorable for subunit addition.

### Prospects for future studies

Our information about how each region of an FH2 domain interacts with the actin filament (*Figure 7* and *Figure 7—figure supplement 1*; *Tables 1* and *2*) can be used to design mutations for experiments to understand mechanisms. For example, one might investigate gating by mutating conserved FH2 residues that form salt bridges with actin. Candidate residues include residues that form salt bridges with R147 of the A2 subunit: E1093 of Cdc12 (99.4% occupied), E1463 of Bni1 (83.6%) and E871 of mDia1 (8.4%) in FHL knobs (*Table 1*). Other candidates for mutation form salt bridges with D363 of the A2 subunit: R990 of Cdc12 (100.0% occupied), K1357 of Bni1 (85.2%), and R764 of mDia1 (23.4%) of the FHL lasso (*Table 2*). However, our studies show that gating involves complicated, global conformational changes that may not be amenable to perturbation by simple mutations. For example, swapping linkers between these three formins changed the lifetime of the chimeric formins on the ends of growing filaments but did not alter gating (*Paul and Pollard, 2009a*). Therefore, structural studies and biophysical methods such as fluorescence resonance energy transfer may be more likely than mutations to provide more information about gating.

## Materials and methods

### System setup for all-atom simulations

Crystal structures of mDia1 and Cdc12 FH2 domains associated with an actin filament are not available, so we generated their homology models by using our structure of a Bni1/actin seven-mer obtained by all-atom (AA) MD simulations (*Baker et al., 2015*) starting with the crystal structure of *Otomo et al. (2005)*. For homology modeling, amino acid sequences of mDia1 and Cdc12 from Uniprot (*The UniProt Consortium, 2017*), and the pdb file of Bni1 (as a template) were supplied to SWISS-MODEL web server (*Kiefer et al., 2009*; *Arnold et al., 2006*) (RRID:SCR_013032). MODELLER (*Eswar et al., 2006*; *Fiser et al., 2000*) (RRID:SCR_008395) was used to obtain the structures of missing residues based on ab initio methods. Both resulting Cdc12 FH2 subunits (FHT and FHL) have 402 residues consisting of residues 984 to 1385 and both resulting mDia1 FH2 subunits have 397 residues ranging from residue 754 to 1150. After running minimization, heating and equilibrium dynamics on the homology models of the Cdc12 and mDia1 FH2 domains, we formed the actin-formin complex by incorporating an actin filament composed of seven subunits into the formin dimers. This was achieved by alignment of the formins onto the Bni1-actin complex by using the MultiSeq and Stride tools of VMD (*Humphrey et al., 1996*) (RRID:SCR_001820). The definitions of features of FH2 domains are listed in *Supplementary file 1*. The accuracy of the homology models was assessed via ProSA (Protein Structure Analysis Tool) (*Figure 2A*) to calculate z-scores using knowledge-based potentials of mean force (*Wiederstein and Sippl, 2007*). The z-score is obtained by calculating the deviation in the total energy of the target protein structure from an energy distribution of random protein structures on the Protein Data Bank (PDB). To confirm the quality of the models, we generated various structures of FH2 domains interacting with the barbed-end of an actin filament using template-based homology modeling (Bni1 based on Cdc12, Cdc12 based on mDia1, and mDia1 based on Cdc12). All-atom simulations of these structures were run for about 95 ns to refine these models, which we compared with the crystal structure of Bni1 by using the template modeling score (TM-score) (*Zhang and Skolnick, 2005*). All these structures were found to have a TM-score between 0.5 and 1.0 (*Supplementary file 2*), which indicates that they have about the same fold as the crystal structure of Bni1 (*Zhang and Skolnick, 2005*; *Xu and Zhang, 2010*). As the homology models were compared with the crystal structure of Bni1, the homology model of Bni1 FH2 had the highest TM-score (0.79) consistent with the expected quality of the homology models. Thus, template-based

homology modeling provides protein models with a high degree of similarity in their topological organization of the secondary structures to the original crystal structure.

The initial structures of mDia1 and Cdc12 FH2 domains bound to the barbed-end of an actin filament (*Figure 2B*) were generated by aligning their FH2 domains with the final equilibrated structure of the Bni1/actin 7-mer after 160 ns of AA MD simulations (*Baker et al., 2015*). Then, the FH2–actin complexes of Cdc12 and mDia1 were solvated and ionized using exactly the same procedure to obtain the final configuration of the Bni1 FH2–actin complex. The systems were then solvated by leaving enough distance in each direction (at least 15 Å) to prevent the protein from interacting with its periodic image during the simulations. The N-terminus of the actin monomers was acetylated, and all other residues in the system were modeled in their standard states of protonation at a pH value of 7. The systems were neutralized with 0.18 M KCl. The Cdc12/actin and mDia1/actin systems were energy minimized by gradually releasing the restraints on the systems via the use of NAMD (*Phillips et al., 2005*) (RRID:SCR_014894) with the Charmm27 force field with CMAP corrections (*Mackerell et al., 2004*). In the heating phase, the temperatures of the systems were gradually raised to 310 K during the simulation spanning 200 ps with restraints on the backbone atoms of the protein, ADP, the $Mg^{2+}$ ion and coordinating waters. The heating phase was followed by an equilibration phase in which the restraints on the protein backbone, ADP, the $Mg^{2+}$ ion and coordinating waters were gradually reduced from 10 kcal/mol/Å$^2$ to 0.1 kcal/mol/Å$^2$ over 400 ps. The AA MD simulations of the Cdc12/actin and mDia1/actin systems with unrestrained dynamics were run for 500 ns. The AA simulation of the Bni1 FH2/actin system was extended from 160 ns (*Baker et al., 2015*) for 340 ns to reach 500 ns of dynamics.

## System setup for coarse-grained simulations

To capture the longer time scale dynamics of FH2 dimers on the end of an actin filament, we created CG models using essential dynamics coarse graining (ED-CG) (*Zhang et al., 2008*) and heterogeneous elastic network modeling (heteroENM) (*Lyman et al., 2008*). EDCG is a systematic method that defines CG sites based on the collective protein motions, which are computed using principal component analysis (PCA) of all-atom trajectories. The CG sites of all regions in formin and actin were automatically determined by the EDCG method. Each subunit in the complex consists of 45 CG sites, which correspond to approximately eight and nine residues per CG site, for FH2 and actin monomers. This CG level of resolution was chosen to obtain a model that is both computationally efficient and capable of incorporating spatial structural elements of the complex. The assignment of CG sites was done separately for each subunit (Formin Homology Leading (FHL), Formin Homology Trailing (FHT), and actin subunits A1 to A7, numbered from the barbed end) to capture characteristic fluctuation profiles of individual subunits.

After the CG site assignment stage, intramolecular interactions of formin and actin were determined by using heteroENM. In heteroENM, CG sites within a certain cutoff distance are connected via effective harmonic springs, each with a specific spring constant, all collectively determined iteratively from the AA MD data to match the actual protein fluctuations observed in the AA simulations. The cutoff distances were chosen to obtain the best match between root mean square fluctuations (RMSF) of all-atom and CG simulations (50 Å (FH2) and 30 Å (actin) for Cdc12-actin, 60 Å and 20 Å for Bni1-actin, and 60 Å and 30 Å for mDia1-actin complexes, respectively).

Intermolecular interactions between FH2 and actin were described by 12–6 Lennard Jones (LJ) potentials and screened Coulomb potentials to effectively introduce dispersion and electrostatic forces into the system. According to the Derjaguin-Landau-Verwey-Overbreek theory, these interactions can be assumed to be additive and independent of each other, therefore intermolecular interactions can be written as follows:

$$U_{inter} = U_{LJ} + U_{coulomb} \tag{1}$$

The dispersion and repulsive forces due to excluded volume were modeled using a 12–6 LJ potential (2) and electrostatic forces were introduced by using a screened Coulomb potential (3):

$$U_{LJ} = 4\varepsilon_{ij} \left[ \left( \frac{\sigma_{ij}}{r} \right)^{12} - \left( \frac{\sigma_{ij}}{r} \right)^6 \right] \qquad r < r_{cut} \tag{2}$$

$$U_{\mathrm{Coulomb}} = \frac{q_i q_j}{4\pi\varepsilon_r} \exp(-\kappa r) \tag{3}$$

The repulsive part of the 12–6 LJ potential was used to describe the excluded volume interactions. Epsilon parameters of the LJ potential were chosen to be 10 kcal/mol, and sigma parameters were obtained by calculating the radius of gyration of the CG sites. The cut-offs for the LJ potential were set to the sigma parameters, whereas that of the screened Coulomb potential was set to 30 Å. The charge fitting method was used for assigning a charge to individual CG sites in order to match the charge distribution of the all-atom model (*Baker et al., 2015*). The Debye length ($\kappa$) in the screened Coulomb potential was chosen as 0.8 nm to mimic the physiological salt conditions. There is no consensus on the dielectric constant at the protein-protein interface, but the value is low and depends on the system. Implicit solvent MD simulations (*Brooks et al., 1983*) and molecular mechanics Poisson-Boltzmann (MMPB) methods (*Gilson and Honig, 1991*) commonly consider the dielectric constant to be $\geq 2.0$, whereas continuum methods commonly use a dielectric constant of 4 and even larger for p$K_a$ calculations (*Gilson and Honig, 1988*).

The initial configurations of the CG systems correspond to the structures obtained after 200 ns of all-atom simulations. The CG simulations of all three systems were run with the LAMMPS MD software package (*Plimpton, 1995*) for two microseconds using a temperature of 310° K in the constant *NVT* ensemble under Langevin dynamics (with a 1000 fs damping parameter).

## System setup for metabasin metadynamics simulations

The FH2 domains interacting with five-mer filaments were used as initial structures for metabasin metadynamics (MBMetaD) simulations (*Dama et al., 2015*). These structures were generated by removing actin subunit A1 from the barbed end and the last actin subunit (A7) from the pointed end of the seven-mer filament at the end of the first 200 ns of AA simulations, and they correspond to conformations before the addition of subunit A1. The structures were simulated in GROMACS 5.1.4 (*Van Der Spoel et al., 2005*) (RRID:SCR_014565) patched with a customized version of PLUMED 2 (*Tribello et al., 2014*) using the CHARMM22/27 force field with CMAP (*Mackerell et al., 2004*). The MBMetaD simulations were run with initial hill heights of 0.001 kcal/mol and widths of 0.03 nm deposited every 50 fs. A bias factor of 10 was used to systematically adjust the hill heights based on the well-tempered MetaD rule. The MBMetaD domains were defined to have an energy offset of 6 kcal/mol and updated every 100 ps, if the exterior bias had changed by 0.5 kcal/mol. Two independent replicates of 80 ns were performed for each structure.

## Acknowledgements

This research was supported by the National Science Foundation (NSF) Materials Research Science and Engineering Center (MRSEC) under grant DMR-14207090 (FA and GAV) and by the National Institute of General Medical Sciences of the National Institutes of Health under award numbers R01GM026338 (NC and TDP) and R01GM122787 (NC). The content is solely the responsibility of the authors and does not necessarily represent the official views of the National Institutes of Health. The computational resources in this work were provided in part by the Research Computing Center (RCC) at The University of Chicago and in part by the NSF Extreme Science and Engineering Discovery Environment (XSEDE). These computations were also performed in part by a grant of computer time from the U.S. Department of Defense (DOD) High Performance Computing Modernization Program at the Engineer Research and Development Center (ERDC) and Navy DOD Supercomputing Resource Centers.

## Additional information

### Funding

| Funder | Grant reference number | Author |
| --- | --- | --- |
| National Science Foundation | Materials Research Science and Engineering Center DMR-14207090 | Fikret Aydin Gregory A Voth |

| National Institute of General Medical Sciences | R01GM026338 | Naomi Courtemanche Thomas D Pollard |
| National Institute of General Medical Sciences | R01GM122787 | Naomi Courtemanche |

The funders had no role in study design, data collection and interpretation, or the decision to submit the work for publication.

### Author contributions

Fikret Aydin, Conceptualization, Formal analysis, Validation, Investigation, Visualization, Methodology, Writing—original draft; Naomi Courtemanche, Conceptualization, Formal analysis, Investigation, Writing—review and editing, Funding acquisition; Thomas D Pollard, Conceptualization, Formal analysis, Supervision, Funding acquisition, Investigation, Project administration, Writing—review and editing; Gregory A Voth, Conceptualization, Resources, Formal analysis, Supervision, Funding acquisition, Investigation, Project administration, Writing—review and editing

### Author ORCIDs

Fikret Aydin (iD) http://orcid.org/0000-0003-3237-8043
Thomas D Pollard (iD) http://orcid.org/0000-0002-1785-2969
Gregory A Voth (iD) http://orcid.org/0000-0002-3267-6748

### Decision letter and Author response

Decision letter https://doi.org/10.7554/eLife.37342.031
Author response https://doi.org/10.7554/eLife.37342.032

# Additional files

### Supplementary files

• Supplementary file 1. Definitions of features of Cdc12, Bni1 and mDia1 FH2 domains. The columns show feature name and residues comprising corresponding feature.
DOI: https://doi.org/10.7554/eLife.37342.025

• Supplementary file 2. Template modeling scores (TM-scores) of the FH2 domain homology models based on different templates. TM-score is used to determine the similarity of protein structures. Based on statistics, TM-score between 0.0 and 0.17 indicates random structural similarity and TM-score between 0.5 and 1.00 indicates having about the same fold [43]. TM-scores were obtained through this website: https://zhanglab.ccmb.med.umich.edu/TM-align/
DOI: https://doi.org/10.7554/eLife.37342.026

• Supplementary file 3. Correlations between interactions of FH2 domain with an actin filament and the barbed end configuration. Pearson correlation coefficients for the number of contacts between the lasso, knob and post regions of the FH2 domains and actin subunits (A2 and A3) and the distributions of twist angles of A2-A3 (given in *Figure 5—figure supplement 2* and *Figure 5—figure supplement 1*) during 350 ns all-atom MD simulations of five-mer filaments.
DOI: https://doi.org/10.7554/eLife.37342.027

• Supplementary file 4. Average interactions of different FH2 regions with an actin filament. The average number of contacts (with standard deviations and t-statistics) between the lasso, knob and post regions of the FH2 domains and actin subunits (A2 and A3) from 350 ns of the all-atom MD simulations of five-mer filaments.
DOI: https://doi.org/10.7554/eLife.37342.028

• Transparent reporting form
DOI: https://doi.org/10.7554/eLife.37342.029

### Data availability

Some parts of data generated or analysed during this study are included in the manuscript and supporting files.

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
