## [Decision Letter]

[Editors’ note: a previous version of this study was rejected after peer review, but the authors submitted for reconsideration. The first decision letter after peer review is shown below.]

Thank you for submitting your work entitled "The role of intermolecular interactions in the gating mechanism of formins" for consideration by *eLife*. Your article has been reviewed by three peer reviewers, and the evaluation has been overseen by a Reviewing Editor and a Senior Editor. The reviewers have opted to remain anonymous.

Our decision has been reached after consultation between the reviewers. Based on these discussions and the individual reviews below, we regret to inform you that your work will not be considered further for publication in *eLife*.

The reviewers appreciated the importance of your modeling work and its potential to significantly advance the mechanistic understanding of formin activity. However, they also pointed out ambiguities in interpreting the effect of FH2-actin contacts, actin twist, and steric hindrance based on the provided data. The relation of the results of this work to prior proposed stepping mechanisms and open/close gating was not clear. The reviews also pointed out that the coarse-grained model suggests larger FH2 motions compared to the all atom simulations, and thus both models cannot be used as independent evidence for the same effect. Since it is a policy of *eLife* to invite revisions only when the path to successfully addressing the reviewer comments is clear and the revisions can be completed within two months, and this does not seem to be the case here, we cannot further consider the current version of your paper for publication. However, we will be prepared to consider a new submission, which would fully address the comments of all three reviewers. In case you decide to prepare such a new submission, please note that to resolve the apparent inconsistency between the two types of models currently included in the paper, one of the models could be abandoned in favor of the most accurate and relevant one.

Reviewer #1:

The authors describe several models for the interaction of the formin FH2 domain with actin filaments (all atom MD, metadynamics, coarse grained). They argue that mDia1-bound barbed ends become more accessible for polymerization and Cdc12-bound barbed ends less accessible, compared to Bni1-bound ends. Since this is the same trend as in the experimentally measured gating factors, the authors propose that their simulations provide an explanation of the origin of gating.

Demonstrating that the gating factors of formins can be calculated from first principles would indeed be a great theoretical advance worthy of a publication in *eLife*. It would demonstrate the power of the modeling methods that would have broad implications in the cytoskeleton field and beyond.

However, I have the following comments and concerns:

* The paper will have stronger impact if the authors provide predictions for future experiments based on their improved understanding of the system. For example mutations that could change the gating factor or predictions for the gating factors of other formins that have not yet been measured.

* This study provides many different pieces of evidence pointing towards the picture of gating factors mDia1>Bni1>Cdc12, on average. These studies are very useful given that very little is known at this level. However, because of fluctuations, need for long simulation times and differences between the models make each measure (or their combination) somewhat ambiguous. One such example is the evolution of the twist angles in Figure 4. They show a trend, which is however not completely clear due to fluctuations and lack of clear equilibration. Further, in this case the order seems to be different: mDia1>Cdc12>Bni1. I provide some further examples below.

* Figure 3 demonstrates the volume fraction in two different FH2 configurations at the barbed end that should differ by a step of one FH2 member forward or backward. Presumably only one of the two is the main conformation prior to polymerization of a new actin subunit. I understand there are no stepping fluctuations, which would require a different analysis. If the primary configuration is that in Figure 1A, then the evidence of different excluded volume fractions between formins is rather weak, based on just the last 50 ns or less of the simulation (I may be missing something but there seem to be another 100 ns of these simulations in Figure 4. Shouldn't they be included in Figure 3?). The difference between formins is more clear in the configuration of Figure 1D, however this may not be primary configuration.

* A transient change of the system is to be expected each time a new formin FH2 is simulated. Given that the blocked volume fraction is relatively small for the all atom simulations in Figure 3, it appears to me that studies of the relative motion of just 2 formins with respect to Bni1 may not be sufficient to establish a completely convincing trend (to simplify the argument, it's 25% chance to get two heads, tossing a coin twice).

* Since the steric hindrance effect in the AA MD simulations is relatively small, one cannot be certain on its effect on actin polymerization kinetics. A steric hindrance blocking is a likely possibility but it's also conceivable that incoming actin monomers interact with the FH2 domain in a way that guides them to the barbed end. These aspects are not studied in this paper.

* It's not clear that the three models (all atom MD, metadynamics, coarse grained) provide a coherent story. The magnitude of the steric hindrance effect as well as the FH2 motions calculated by the CG simulations is much larger compared to the AA MD simulations. This seems to suggest that either the AA MD simulations need more time to explore a larger region of space or else that the CG simulations provide unphysical motions and thus cannot be used to calculate the steric hindrance effect. It's also not clear to me why the metadynamics studies were not used to calculate the effect of steric hindrance and actin helical twist.

* The timescale of 200 ns in Figure 1 appears to be too short to reliably predict secondary structure formation at the lasso, post and knob regions. For example, α helices typically take 50-100 ns to form in MD simulations and higher order structures may take even longer.

Reviewer #2:

This manuscript by Aydin and coworkers presents results from molecular dynamics simulations, addressing the conformations of formin FH2 dimers and actin subunits at/near the barbed ends, particularly focusing on barbed end "gating", i.e. the FH2 dimer's ability to allow or prevent the addition of a new actin subunit. The authors find that three factors contribute to gating, with different amplitudes for different formin isoforms: steric interference, flattening of the actin helix, and the strength of the interaction between FH2 and the barbed end.

While I disagree with the authors' argument that "In contrast to gating, much is already known about the transfer of profilin-actin from binding sites on FH1 domains to open barbed ends" (in their rebuttal of the "informal review") I do agree that gating is an important property that needs to be better understood. I also agree that MD simulations are a valuable method to do so, and I find their results interesting. However, I think the authors could in general better present their results in the context of existing models. As they are, their results are often difficult to connect to what we currently understand about formin activity.

I think this work should be of interest to readers beyond MD simulations specialists, if the authors improve their manuscript by addressing the following points.

1) Two main models are currently available (and often used to understand experimental data – see for example the recent Kubota et al., 2017) to describe FH2 gating and filament elongation: the so-called "stair-stepping" and "stepping second" models. They are extensively presented in a series of papers by Paul and Pollard from 2008-2009. It is frustrating that the current manuscript does not provide any additional insight on this question, which seems central here.

Can the present simulations shed light on these models? Do they favor one model over the other?

Hemidimer translocation is never explicitly mentioned. Is it nonetheless included in the present simulations, and if not, how would one integrate it?

2) I understand that, as written in the last sentence of the manuscript, the conformations fluctuate rapidly. But what time scales are we talking about? Are the fluctuations shown in this manuscript large enough to account for the equilibrium between open and closed states, or should we expect these transitions to take place over larger time scales?

3) Fluctuations and the FH2 dimer's ability to explore different conformations are addressed more explicitly in Figure 6 and in the subsection “Conformational mobility of FHT domains is consistent with the strength of intermolecular interactions”, where it is written that "Cdc12 FHT domain explores a smaller area than Bni1 and mDia1 FHT domains". However, these results seem in contradiction with those of Figure 7, where the FHT of Cdc12 has larger fluctuations and larger displacements than mDia1 and bni1. Please clarify.

4) In the Introduction, the gating factor is defined as the "fraction of the time that the FH2 domains are found in the open state". This is assumed to be the same as the gating factors determined as the ratio of the formin elongation rate to the free barbed end elongation rate (in the absence of profilin). However, equating these two definitions relies on one important hypothesis: that the on-rate constant for monomer addition is the same for a free barbed end and for an open-state formin barbed end. This should be specified. It should also be discussed in light of the present results: does this hypothesis appear valid, now that we know more about the conformations adopted by the filament in interaction with an FH2 dimer?

5) Gating is a schematic, all-or-nothing frame to work in. For instance, the open and closed states are generally defined by the 167° versus 180° twists of the filament. These extreme angles do not appear to be reached in the simulations, which thus seem to describe intermediate situations. Please comment. For example, should we rather consider a continuum of states, with different on-rate constants for the addition of actin monomers?

6) The simulations do not impose any constraint (boundary conditions) on the last subunits (A6 and A7 in the seven-mer) which thus correspond to the free pointed ends of very short filaments. The results (e.g. Figure 4 on twist angles) indicate that changes in conformation propagate over several subunits. One would then expect different results for longer filaments, which could perhaps be simulated by imposing the canonical filament conformation as a boundary condition. Please comment on this limitation of the model. How would the results extrapolate to longer filaments?

7) FH2 dimer detachment from barbed ends (Figure 8) is quite puzzling. What should one make of this? Do the authors expect this result to correspond to formin detachment from barbed ends (which appears to have a very low off-rate constant in experiments) or does it illustrate a limitation of the coarse grain method?

Reviewer #3:

Overall, the goal is of this paper, to use MD to test models of formin gating is an important one. However, the approach and presentation leave a lot to be desired. The topic is suited for a broader audience but the paper is not written with a broad audience in mind. Connecting the data to the model, simply in terminology, would have helped. Is the structure with a 5mer of actin expected to relax into a "closed" state? Is the structure with the 7mer of actin forced into an "open" state? This brings up questions about steric clashes observed when AA MD calculations were performed – Shouldn't the clashes be disallowed?

In general, the data are presented as lists, leaving it to the reader to interpret everything and/or guess how the authors are interpreting the data. My read doesn't agree with the final conclusions of the authors but I can't figure out why. For example, when looking at contacts, Bni1 was often more like Cdc12, which fits with the published gating data. However, when considering actin orientation, Bni1 was more like mDia1. Yet, the authors conclude that their data agree well with experimental gating data and that both steric clashes and actin twisting play a role in gating.

The paper was originally criticized for being entirely computational. The authors counter that the data correlate nicely with "wet" data in the literature but the correlations aren't as clean as the authors would have us believe. Further, they remain merely correlations until someone demonstrates that a manipulation to the proteins (in vitro and/or in silico) produces the predicted results. Along these lines, some of the work felt circular. It might have helped if the authors used the other co-crystal published (FMNL/actin) in addition to the Bni1/actin structure.

A second criticism was about focus on elongation/gating, which was not a concern for me. However, the authors state the FH1 domains can overcome the effect of gating but this study is done in the absence of FH1 domains, raising the question of how valuable it is. This is a point that needs to be addressed.

---

## [Author Response]

[Editors’ note: the author responses to the first round of peer review follow.]

Reviewer #1:[…] I have the following comments and concerns:* The paper will have stronger impact if the authors provide predictions for future experiments based on their improved understanding of the system. For example mutations that could change the gating factor or predictions for the gating factors of other formins that have not yet been measured.

Thanks for the suggestion. We added the following to the Discussion:

“Our information about how each region of an FH2 domain interacts with the actin filament (Figures 7 and Figure 7—figure supplement 1; Tables 1 and 2) can be used to design mutations for experiments to understand mechanisms. […] Therefore, structural studies and biophysical methods such as fluorescence resonance energy transfer may be more likely than mutations to provide more information about gating.”

Another part of the Discussion notes that our results suggest that Cdc12 is the best nucleator of the three formins.

* This study provides many different pieces of evidence pointing towards the picture of gating factors mDia1>Bni1>Cdc12, on average. These studies are very useful given that very little is known at this level. However, because of fluctuations, need for long simulation times and differences between the models make each measure (or their combination) somewhat ambiguous. One such example is the evolution of the twist angles in Figure 4. They show a trend, which is however not completely clear due to fluctuations and lack of clear equilibration. Further, in this case the order seems to be different: mDia1>Cdc12>Bni1. I provide some further examples below.

The reviewer is correct that the simulation in old Figure 4 produced twist angle distributions different from the experimental ranking of gating factors (mDia1>Bni1>Cdc12). To investigate further, we continued the AA MD simulations of seven-mer filaments out to 500 ns and found that the twist angles of terminal actin subunits associated with the mDia1 FH2 were close to 167° while those associated with Cdc12 or Bni1 FH2 fluctuated mostly in the angle range of 169- 173° (new Figures 4E, G), more in keeping with the gating factors.

Two new simulations of five-mer filaments, the state prior to addition of the new subunit, showed that the most probable twist angles were 168° and 170° for mDia1, 172° for Bni1 and 174° and 175° for Cdc12 (Figures 5B, C) with fluctuations for each over a range of about 4°. Therefore, mDia1 FH2 favors the ‘open’ configuration of barbed ends awaiting the association of a subunit, while Cdc12 favors twist angles that compromise subunit addition.

Furthermore, during two new metadynamics simulations of five-mer filaments (Figure 6C), the twist angles of filaments with Cdc12 and mDia1 differed the most, as Bni1 varied between them, again in line with the gating factors.

We revised the manuscript to emphasize that new Figure 4 shows twist angles after addition of a new actin subunit and before stepping of the trailing FH2 domain onto the new subunit in preparation for addition of the next subunit. We suggest that the new subunit might limit the flattening of the barbed end by the formins.

* Figure 3 demonstrates the volume fraction in two different FH2 configurations at the barbed end that should differ by a step of one FH2 member forward or backward. Presumably only one of the two is the main conformation prior to polymerization of a new actin subunit. I understand there are no stepping fluctuations, which would require a different analysis. If the primary configuration is that in Figure 1A, then the evidence of different excluded volume fractions between formins is rather weak, based on just the last 50 ns or less of the simulation (I may be missing something but there seem to be another 100 ns of these simulations in Figure 4. Shouldn't they be included in Figure 3?). The difference between formins is more clear in the configuration of Figure 1D, however this may not be primary configuration.

The FH2 domains interacting with seven-mer filament correspond to the configuration just after addition of a new actin subunit, whereas the configuration in the five-mer filament is before addition of a subunit. Therefore, we expect that FH2 domains interacting with the five-mer filament to have configurations more relevant to investigate the mechanisms affecting the addition of a new actin subunit and they are more relevant to the experimentally observed differences in the polymerization rates of actin by three formins.

In response to the reviewer’s suggestion, we extended the length of all-atom simulations in the revised paper, obtaining new data that support our conclusions. During the longer simulations of seven-mer filaments (extended from 200 to 500 ns), the steric interference of FHL with subunit +A1 continued to fluctuate at high levels for Cdc12, fluctuated between none to low levels for mDia1, and fluctuated at intermediate levels for Bni1 (new Figure 3A).

Extension of the simulations of five-mer filaments from 125 ns to 350 ns showed that the steric interference of FHT with subunit A1 fluctuated at higher levels for Cdc12 and fluctuated between none and low levels for Bni1 and mDia1 (new Figure 3D). Additional 200 ns, replicate simulations of five-mer filaments also demonstrated similar differences in the steric interference of the three formin FHT domains with subunit A1 (new Figure 3E).

* A transient change of the system is to be expected each time a new formin FH2 is simulated. Given that the blocked volume fraction is relatively small for the all atom simulations in Figure 3, it appears to me that studies of the relative motion of just 2 formins with respect to Bni1 may not be sufficient to establish a completely convincing trend (to simplify the argument, it's 25% chance to get two heads, tossing a coin twice).

We addressed this concern by running longer all atom simulations in new Figures 3A, D and observed larger blocked volume fractions. We also replicated the key experiments (Figure 3E). The additional simulations showed similar trends to the previous simulations (note that the second replicate is shorter than the first replicate), as explained in the previous response. The revised text emphasizes that both steric blocking and flattening of the barbed end contribute to impeding elongation, so neither stands alone.

* Since the steric hindrance effect in the AA MD simulations is relatively small, one cannot be certain on its effect on actin polymerization kinetics. A steric hindrance blocking is a likely possibility but it's also conceivable that incoming actin monomers interact with the FH2 domain in a way that guides them to the barbed end. These aspects are not studied in this paper.

Even a small steric clash will block binding of a new actin subunit. The reviewer proposes that incoming actin subunits might interact with a FH2 domain and somehow overcome steric interference. We are not aware of any evidence supporting this fascinating, induced-fit hypothesis.

* It's not clear that the three models (all atom MD, metadynamics, coarse grained) provide a coherent story. The magnitude of the steric hindrance effect as well as the FH2 motions calculated by the CG simulations is much larger compared to the AA MD simulations. This seems to suggest that either the AA MD simulations need more time to explore a larger region of space or else that the CG simulations provide unphysical motions and thus cannot be used to calculate the steric hindrance effect. It's also not clear to me why the metadynamics studies were not used to calculate the effect of steric hindrance and actin helical twist.

We agree that the experiments in the original version of our manuscript did not provide a coherent story. Fortunately, new experiments and analysis have clarified the picture.

First, a repeated the metadynamics simulations confirmed the original findings (new Figure 6A). At the outset of these MB-MetaD simulations the steric clashes between the FHT do mains of the three formins and the incoming actin subunit A1 differed for the three formins. The FHT domains must rearrange to escape from these steric clashes. The energy landscapes in Figure 6A show that the FHT domain of Cdc12 is more confined by high energy barriers than Bni1 or mDia1. These high energy barriers for conformational rearrangements restrict the FHT domain of Cdc12 to a smaller area of the conformational space (including those with steric clashes with A1) than the FHT domains of Bni1 and mDia1.

Following the reviewer’s suggestion, we also investigated the twist angle distributions of three formins during these simulations. Although these simulations are much shorter than the AA simulations (80 ns) and the helical twist is not biased, we observed a similar trend: largest difference exists between Cdc12 and mDia1, and Bni1 varies between them (new Figure 6C). The twist angles distribution of Bni1 was found more similar to that of mDia1 than Cdc12, which is consistent with their gating factors. The metadynamics simulations also revealed a correlation between the conformational mobility of formins and their effects on distorting the barbed end of actin filament. The Cdc12, which has the lowest conformational mobility, distorted the barbed end more than Bni1 and mDia1.

The issue with the CG simulations was that the FH2 domains dissociated from the barbed end of five-mer filaments when the dielectric constant was 5 and even lower (old Figure 8). In the revised manuscript, we compared the behavior of seven-mer filaments during CG simulations with dielectric constants of 1 to 3 to change the strength of electrostatic interactions at the protein-protein interface. With ε = 1, the three formins deviated only slightly from their initial configurations (Figure 8E). With ε = 3, the Cdc12 FH2 domain underwent larger fluctuations than Bni1 and mDia1 FH2 domains (Figure 8F). This behavior is consistent with our all-atom simulations showing more salt bridges between Cdc12 and the actin filament than Bni1 and mDia1 (Table 1). Thus, electrostatic interactions are more important for Cdc12 than Bni1 and mDia1. In addition, the CG simulations showed that the largest conformational difference exists between Cdc12 and mDia1, and Bni1 varies between them at high dielectric constant. This trend is similar to the trends observed in the AA simulations (such as results on the steric hindrance and twist angle distributions), in which the largest differences were found between Cdc12 and mDia1, and Bni1 varied between them.

Although barbed ends with FH2 domains simply fluctuated around their equilibrium configurations without flattening during the CG simulations (owing to the effective harmonic bonds between the subunits in heteroENM), these simulations (Figure 9) provided information on steric blocking that complements the AA simulations (Figure 3). During 2 µs CG MD simulations the center of mass of the Cdc12 FHL domain shifted radially toward the filament axis where it interferes with the incoming actin subunit (Figure 9C). The FHL domains of Bni1 and mDia1 fluctuated around their initial positions where intermittent steric blocking was observed in the AA simulations (Figure 3).

Given that the CG simulations are less important than the AA and metadynamics simulations for understanding gating, we used two figures (new Figures 8, 9) to present CG data rather than three in the original submission.

* The timescale of 200 ns in Figure 1 appears to be too short to reliably predict secondary structure formation at the lasso, post and knob regions. For example, α helices typically take 50-100 ns to form in MD simulations and higher order structures may take even longer.

Our aim is not to observe formation of the secondary structures during MD simulations. We generated homology models and used AA MD simulations to refine these models. The RMSD plots (Figure 2D) show that the structures stabilized before 200 ns, after which we assessed conformational differences.

Reviewer #2:[…] I think this work should be of interest to readers beyond MD simulations specialists, if the authors improve their manuscript by addressing the following points.1) Two main models are currently available (and often used to understand experimental data – see for example the recent Kubota et al., 2017) to describe FH2 gating and filament elongation: the so-called "stair-stepping" and "stepping second" models. They are extensively presented in a series of papers by Paul and Pollard from 2008-2009. It is frustrating that the current manuscript does not provide any additional insight on this question, which seems central here.Can the present simulations shed light on these models? Do they favor one model over the other?

Thank you for this interesting question, which has several dimensions:

1) Since processive association of an FH2 dimer with the end of the growing filament depends on addition of actin subunits and since gating limits addition of actin subunits, our new results on gating can be applied to both models of elongation. The all atom and metadynamics simulations revealed two underlying mechanisms, steric interference or flattening of the barbed end, that contribute to formins slowing actin filament elongation and scale with the gating factors. Extensive new experiments and analysis in the revised manuscript strengthen this conclusion. Regardless of whether the FHT domain steps first or second, these factors will limit the access of new subunits to the end of the filament.

2) The simulations confirmed the original idea from the stair-stepping model that FH2 domains can sterically block a barbed end [Pring et al., 2003] but revealed that thermal motions can move an FH2 domain out of a blocking site to a conformation that allows addition of a new actin subunit. In fact, the energy barriers for these local adjustments are expected to be much lower than for complete dissociation of an FH2 domain, as proposed in “stair-stepping” models of processive elongation. This energetic consideration argues against stair-stepping models.

3) Our simulations focused on gating do not include large scale conformational changes (such as FH2 translocation or stepping) or actin subunit addition. We agree with the reviewer that more mechanistic information about actin subunit addition from solution or by transfer from an FH1 domain would be interesting and valuable to the field. However, as noted by reviewer 1, a different approach would be required to study these important reactions.

Hemidimer translocation is never explicitly mentioned. Is it nonetheless included in the present simulations, and if not, how would one integrate it?

We did not model stepping of FHT onto the new subunit A1, because simulating this reaction at an all-atom level is computationally prohibitive due to the large time and length scales. We revised the beginning of the Discussion to explain these limitations.

2) I understand that, as written in the last sentence of the manuscript, the conformations fluctuate rapidly. But what time scales are we talking about? Are the fluctuations shown in this manuscript large enough to account for the equilibrium between open and closed states, or should we expect these transitions to take place over larger time scales?

Thanks for this question. In response we added a new section to the Discussion entitled “How do the equilibrium conformations of formins on barbed ends fluctuate in time?”, which makes the following points.The fluctuations in our simulations occur in nanoseconds. As mentioned in one of the reviewer’s next comments, these formin-actin complexes rarely visit extreme conformations (such as twist angles of 167° or 180°) during our nanosecond simulations, but we observe transitions in between these angles, therefore the results at nanoscale can indicate the behavior at the longer time scales. As we extended our all-atom simulations an additional 200-300 ns, the barbed end associated with Cdc12 started to visit angles around 178°. Although the probability of these extreme angles was very low in the extended simulations, these simulations showed that these low probability, high energy states can be reached during long simulations. On a millisecond time scale (a few thousand times longer than current simulations) the barbed end interacting with each formin will visit both the open and closed states.

3) Fluctuations and the FH2 dimer's ability to explore different conformations are addressed more explicitly in Figure 6 and in the subsection “Conformational mobility of FHT domains is consistent with the strength of intermolecular interactions”, where it is written that "Cdc12 FHT domain explores a smaller area than Bni1 and mDia1 FHT domains". However, these results seem in contradiction with those of Figure 7, where the FHT of Cdc12 has larger fluctuations and larger displacements than mDia1 and bni1. Please clarify.

The data in Figure 6 from metabasin metadynamics simulations explore different conformations in FH2 domains (different regions of FH2 domains can extend and compress without any restraints). On the other hand, the data in old Figure 7 are from coarse-grained simulations and the conformations of FH2 domains only fluctuate around the equilibrium conformation due to the elastic network. Old Figure 7 only indicates change in the position of FH2 domains with respect to the barbed end of actin filament. The reason we observe large fluctuations for Cdc12 in old Figure 7 is loss of contact between FHT and actin subunit A3 due to the high dielectric constant. We deleted the data from CG simulations with high dielectric constant, because they are confusing and not relevant to the main focus of the paper.

In the revised manuscript we used the CG simulations to test some AA simulation results and to test the effects of electrostatic interactions on FH2 domains at longer times. We tested dielectric constants from 1 to 3 to vary the strength of electrostatic interactions at the protein-protein interfaces. The dielectric constant has the largest effect on Cdc12 FH2 (new Figures 8E, F), consistent with the all-atom simulations showing more salt bridges between Cdc12 and the actin filament than Bni1 or mDia1 (Table 1). These results suggest that the electrostatic interactions are more important during interactions with actin for Cdc12 than Bni1 or mDia1. In addition, the CG simulations showed that the largest conformational difference exists between Cdc12 and mDia1, and Bni1 varies between them at dielectric constant of 3. This trend is similar to the trends observed in the AA results (such as results on the steric hindrance and twist angle distributions), in which the largest differences were found between Cdc12 and mDia1, and Bni1 varied between them.

4) In the Introduction, the gating factor is defined as the "fraction of the time that the FH2 domains are found in the open state". This is assumed to be the same as the gating factors determined as the ratio of the formin elongation rate to the free barbed end elongation rate (in the absence of profilin). However, equating these two definitions relies on one important hypothesis: that the on-rate constant for monomer addition is the same for a free barbed end and for an open-state formin barbed end. This should be specified. It should also be discussed in light of the present results: does this hypothesis appear valid, now that we know more about the conformations adopted by the filament in interaction with an FH2 dimer?

Thanks for this helpful suggestion. We revised the Discussion to include the assumption that collision rate constant is the same for actin monomers interacting with a free barbed end and the open-state of a barbed end with a formin. In response to the reviewer’s insight, we added to the Discussion that the “available experimental measurements cannot distinguish between open states adding subunits with the normal rate constant or a larger population of open states adding subunits slower than free barbed ends.” The discussion of this point goes on to say “Normally the probability that a collision between a diffusing actin monomer and the barbed end of a filament results in binding is high (~2%) [Drenckhahn and Pollard, 1986]. Gating should not change the rate constant for collisions with the end of a filament associated with an FH2 domain, but the probability of binding will be less than 2% depending on the particular formin.”

5) Gating is a schematic, all-or-nothing frame to work in. For instance, the open and closed states are generally defined by the 167° versus 180° twists of the filament. These extreme angles do not appear to be reached in the simulations, which thus seem to describe intermediate situations. Please comment. For example, should we rather consider a continuum of states, with different on-rate constants for the addition of actin monomers?

Thanks again for raising this interesting point. The revised Discussion explains that the formin actin filament complexes undergo continuous thermal motions that allow them to sample a range of conformations including open and closed states, a continuum as suggested by the reviewer. The probability of visiting any state depends on the free energy difference between the equilibrium state and other higher energy states. Extreme states are visited less often than the equilibrium state. Our results show that on a nanosecond time scale each formin favors a different equilibrium state for the barbed end such as Cdc12 favoring larger helical twist angles than Bni1. On time scales relevant to actin filament elongation, barbed ends will visit a wide range of conformations, but the distributions will differ such that barbed ends with Cdc12 will be in closed conformations most of the time, while ends with mDia1 are in open conformations most of the time, but both reach extreme angles a fraction of the time at longer time scales. The section concludes, “Therefore, the gating factor is a time-average of the degree to which the formin compromises subunit addition rather than literally being in ‘open’ or ‘closed’ states.”

6) The simulations do not impose any constraint (boundary conditions) on the last subunits (A6 and A7 in the seven-mer) which thus correspond to the free pointed ends of very short filaments. The results (e.g. Figure 4 on twist angles) indicate that changes in conformation propagate over several subunits. One would then expect different results for longer filaments, which could perhaps be simulated by imposing the canonical filament conformation as a boundary condition. Please comment on this limitation of the model. How would the results extrapolate to longer filaments?

Our simulations show that the strongest effects of FH2 domain on the helical twist are on actin subunits that interact with FH2 domains. Actin subunits close to the pointed end of seven-mer filaments have twist angles close to 167° (Figure 4—figure supplement 1). Thus, the conformation of the free pointed end is similar to a canonical filament without imposing a constraint.

7) FH2 dimer detachment from barbed ends (Figure 8) is quite puzzling. What should one make of this? Do the authors expect this result to correspond to formin detachment from barbed ends (which appears to have a very low off-rate constant in experiments) or does it illustrate a limitation of the coarse grain method?

The high dielectric constant of 5 in our previous CG simulations of five-mers caused all of the FH2 domains to detach from barbed ends, likely an artifact. The revised manuscript focuses on CG simulations of seven-mer filaments, since we parameterized the CG model with AA trajectories of seven-mer filaments. We tested the importance of electrostatic interactions by varying dielectric constant in the CG simulations. The new results with ε = 3 (Figures 8 and 9) support the importance of electrostatic interactions, with the largest effect on the conformation of Cdc12 FH2–actin complex.

Reviewer #3:Overall, the goal is of this paper, to use MD to test models of formin gating is an important one. However, the approach and presentation leave a lot to be desired. The topic is suited for a broader audience but the paper is not written with a broad audience in mind. Connecting the data to the model, simply in terminology, would have helped.

Thanks for sharing your problems with understanding the manuscript. We authors were also concerned that the manuscript was a complicated collection of information without a good story line. Empowered by new data, which clarified the mechanisms, we completely reorganized, rewrote and condensed the manuscript to present the work more clearly for a general audience. We are much happier with the revised figures and text and hope that the reviewer agrees.

Is the structure with a 5mer of actin expected to relax into a "closed" state? Is the structure with the 7mer of actin forced into an "open" state?

Neither structure was forced into an “open” or “closed” state. The structures with five-mer and seven-mer filaments correspond to the states before and after addition of a new actin subunit, respectively. We focus on understanding how the FH2 domains of three formins with different gating factors interact with the barbed ends of actin filament in both of those states. Our aim was to determine the underlying mechanisms (e.g. the barbed-end configuration or steric interference) that slow actin filament elongation.

This brings up questions about steric clashes observed when AA MD calculations were performed – Shouldn't the clashes be disallowed?

Thanks for asking. Of course, steric clashes were not allowed or observed in the MD simulations. After performing the MD simulations, we evaluated if the FH2 domains occupied any positions creating steric clashes with incoming actin subunits.

In general, the data are presented as lists, leaving it to the reader to interpret everything and/or guess how the authors are interpreting the data. My read doesn't agree with the final conclusions of the authors but I can't figure out why. For example, when looking at contacts, Bni1 was often more like Cdc12, which fits with the published gating data. However, when considering actin orientation, Bni1 was more like mDia1. Yet, the authors conclude that their data agree well with experimental gating data and that both steric clashes and actin twisting play a role in gating.

We extended the simulations and collected much additional data that greatly clarified the picture. Both the extent of steric interference and flattening of the actin helix scale with the gating factors and the abilities of the three formins to inhibit elongation, while other parameters including buried surface area and numbers of contacts between the formins and actin do not. These data make it much easier to explain the results in the revised manuscript and to draw multiple firm conclusions in the Discussion.

The paper was originally criticized for being entirely computational. The authors counter that the data correlate nicely with "wet" data in the literature.

The literature already contains extensive experimental information on gating without a mechanistic explanation. Our simulation results provide two mechanisms that can explain the experimental observations and can be used to design better experiments in the future.

The correlations aren't as clean as the authors would have us believe.

We carried out longer all-atom simulations and multiple independent simulations with random initial velocities. The new data (Figures 3, 4, 5, and 6) strengthen the relationship between the experimental gating factors (mDia1 > Bni1 > Cdc12) and the effects of the three types of FH2 domains on the helical twist of the barbed end and on steric clashes with incoming actin subunits.

Further, they remain merely correlations until someone demonstrates that a manipulation to the proteins (in vitro and/or in silico) produces the predicted results. Along these lines, some of the work felt circular.

Our studies demonstrate that gating involves complicated, global conformational changes that are not amenable to perturbation by simple mutations. For example, even swapping linkers between these three formins failed to change gating [Paul and Pollard, 2009]. Therefore, we end the Discussion arguing that structural and biophysical studies are more likely than mutational analysis to advance our understanding of gating.

It might have helped if the authors used the other co-crystal published (FMNL/actin) in addition to the Bni1/actin structure.

Thompson et al., 2013 determined a 3.4 Å crystal structure of a dimer of FMNL3 FH2 domains bound to actin. This structure has a two-fold axis of symmetry between physically separated actin subunits, so it is less informative regarding the structure of FH2 domains bound to a helical actin filament than the Bni1-FH2-actin structure, upon which Baker et al., 2015 and we have based our models for MD simulations. However, the contacts between the FMNL3 FH2 domain and actin are similar to those made by the Bni1-FH2 domains, as we now note in the text with a reference to Thompson et al. We plan a separate study of the FMNL3 FH2 bound to an actin filament.

Mutations of formin FMNL3 had implicated the post and lasso regions in gating: mutations K800A and R782A in the post and R570A in the lasso slowed actin filament elongation, presumably by increasing the gating factor of ~0.4 [1]. During AA simulations of five-mer filaments, flattening of the barbed end by Cdc12 FH2 was strongly correlated with the number of contacts between actin subunits A2 and A3 and the post regions of both FHT and FHL (Figure 5—figure supplement 2 and Supplementary file 3). The sharp increase in the number of contacts for twist angles >173° is also seen as an abrupt change in the time course data in Figure 5D (blue line is contacts between FHL post and A2). This correlation does not establish causality, but the additional contacts may provide the free energy change for the unfavorable change in the conformation of the filament. On the other hand, the contacts between the Cdc12 FH2 lasso or knob regions of Cdc12 and actin subunits A2 and A3 did not change as the filament flattened (Figure 5—figure supplement 1). During these simulations, neither Bni1 nor mDia1 caused a large change in the twist angles or the numbers of contacts of the knob, lasso or post regions with actin (Figure 5—figure supplements 1 and 3 and Supplementary file 3).

A second criticism was about focus on elongation/gating, which was not a concern for me. However, the authors state the FH1 domains can overcome the effect of gating but this study is done in the absence of FH1 domains, raising the question of how valuable it is. This is a point that needs to be addressed.

Our statement “Formin FH1 domains can overcome the effects of gating and increase the rate of actin elongation” was misleading. The point is that concentrating profilin-actin on FH1 domains and transferring actin rapidly to barbed ends allows filaments to elongate faster, *in spite of gating*. We addressed this comment in the Introduction of the revised manuscript:

“Rapid transfer of actin from FH1 domains onto open barbed ends allows filaments to elongate rapidly, in spite of gating”.